# Premature polyadenylation of *MAGI3* produces a dominantly-acting oncogene in human breast cancer

Thomas K Ni[1,2,3]*, Charlotte Kuperwasser[1,2,3]*

[1]Department of Developmental, Chemical and Molecular Biology, Tufts University, Boston, United States; [2]Molecular Oncology Research Institute, Boston, United States; [3]Tufts Medical Center, Boston, United States

**Abstract** Genetic mutation, chromosomal rearrangement and copy number amplification are common mechanisms responsible for generating gain-of-function, cancer-causing alterations. Here we report a new mechanism by which premature cleavage and polyadenylation (pPA) of RNA can produce an oncogenic protein. We identify a pPA event at a cryptic intronic poly(A) signal in *MAGI3*, occurring in the absence of local exonic and intronic mutations. The altered mRNA isoform, called *MAGI3^pPA*, produces a truncated protein that acts in a dominant-negative manner to prevent full-length MAGI3 from interacting with the YAP oncoprotein, thereby relieving YAP inhibition and promoting malignant transformation of human mammary epithelial cells. We additionally find evidence for recurrent expression of *MAGI3^pPA* in primary human breast tumors but not in tumor-adjacent normal tissues. Our results provide an example of how pPA contributes to cancer by generating a truncated mRNA isoform that encodes an oncogenic, gain-of-function protein.

**\*For correspondence:**
kthomasni@gmail.com (TKN);
Charlotte.kuperwasser@tufts.edu (CK)

**Competing interests:** The authors declare that no competing interests exist.

## Introduction

Genetic mutation, copy number gain/loss and chromosomal rearrangement are major mechanisms of genome alteration that underlie cancer pathogenesis. Frequent alteration of a gene by one or more of these mechanisms represents *prima facie* evidence for its role as a *driver* – a gene whose alterations are causative in cancer. As such, within the last decade, tremendous efforts have focused on the systematic, genome-wide surveys for these events in cancer. Interestingly, results from these analyses indicate that many tumors harbor alterations in only 1–2 known drivers, and ~15% of tumors lack alterations in even a single known driver (*Imielinski et al., 2012*; *Garraway and Lander, 2013*). Since solid tumors arise due to oncogenic cooperation between alterations in multiple drivers (*Fearon and Vogelstein, 1990*; *Hanahan and Weinberg, 2000*), a significant number of cancer-relevant genes are currently being missed by genomic analyses because so many genes are ostensibly altered at frequencies below the threshold for driver detection (*Lawrence et al., 2014*).

While it has been posited that unidentified drivers are difficult to identify because they are infrequently altered in cancer, it is equally plausible that additional mechanisms, beyond those regularly interrogated by current genomics platforms, may be responsible for generating driver alterations. In fact, as new mechanisms of alteration relevant to cancer are discovered, a number of 'infrequently altered' genes are reclassified as frequently altered. For example, the discovery of promoter mutations in the gene encoding telomerase has resulted in the reclassification of *TERT* as a frequently altered melanoma driver (*Huang et al., 2013*; *Horn et al., 2013*). Therefore, identifying cancer-specific alterations generated by previously unappreciated mechanisms represents a fundamentally important prerequisite for driver gene identification.

**eLife digest** Cancer is a disease that is caused by the uncontrolled growth of cells. Normal cells can become cancerous if they acquire genetic mutations that allow them to divide more rapidly and ignore certain growth-halting signals from other cells in the body. Therefore, researchers are studying and cataloguing all the genetic changes found in cancers with the hope that this information will provide a clearer understanding of how they start to develop. Some types of mutations are well studied and easily identified by current technologies. However, some tumors contain few of these well-known mutations, which suggests that other types of mutations may be involved.

Ni and Kuperwasser set out to discover some of these other types of genetic mutations in cancer cells and to find out how they contribute to cells becoming cancerous. Initial experiments showed that human breast cancer cells contain variants of proteins not found in healthy cells. In particular, there was a protein called MAGI3 that was abnormally short, but not due to any well-known mutations. This, in turn, led to the discovery of a type of mutation – called premature polyadenylation – affecting the gene that encodes MAGI3. Next, Ni and Kuperwasser used biochemical techniques to show that the shortened MAGI3 protein inappropriately switches on another protein called YAP, which causes cells to grow more quickly.

Approximately 7.5% of breast cancer patients have the shortened version of MAGI3 due to premature polyadenylation, which indicates that this might be a more widespread mutation than previously thought. Future studies will have to determine exactly how premature polyadenylation occurs in cancer cells, and whether it affects other genes and is found in other types of cancer.

Ni et al. previously identified several dozen candidate cancer driver genes from a forward genetics screen for tumorigenesis in mice (*Ni et al., 2013*). Despite being functionally implicated in tumorigenesis, some of these candidate drivers were rarely mutated, amplified or deleted in human cancer. This incongruence led us to hypothesize that some of these candidate drivers may appear to be infrequently altered because their driver alterations may be caused by mechanisms that are not regularly interrogated. Here, we report that one of these candidate drivers, *MAGI3*, is indeed recurrently altered in breast cancer by premature cleavage and polyadenylation (pPA), a mechanism not widely appreciated for its involvement in cancer, and the resulting truncation of the *MAGI3* gene products functionally contributes to malignant transformation.

## Results

### A premature polyadenylation event generates a truncated MAGI3 protein in MDA-MB-231 breast cancer cells

Many mechanisms are involved in conveying genetic information from genes to their mRNA and protein products. When gone awry, any of these mechanisms could alter cancer-relevant genes or their gene products. To investigate our hypothesis that some candidate drivers may be altered by underlying mechanisms not widely thought to be involved cancer, we considered strategies capable of broadly capturing many types of alteration events. While there is no standard strategy for this purpose, we reasoned that changes in genetic information, in most cases, must ultimately manifest at the protein level in order to contribute to cancer. Thus, our strategy focused on identifying novel, unannotated protein products of a limited number of genes that we suspected to be involved in cancer due to their previous identification by a forward genetics screen for tumorigenesis in mice (*Ni et al., 2013*). We focused specifically on products exhibiting detectable size differences from wild-type protein isoforms since these may be more likely to cause significant functional effects. Once such a product was identified, we would determine whether the altered protein is generated by a genetic mutation in the coding regions of the corresponding gene (*Figure 1A*). If not due to coding region genetic mutation, we would attempt to identify the alteration mechanism, establish recurrence for the specific alteration event in cancer and investigate its functional importance in malignant transformation.

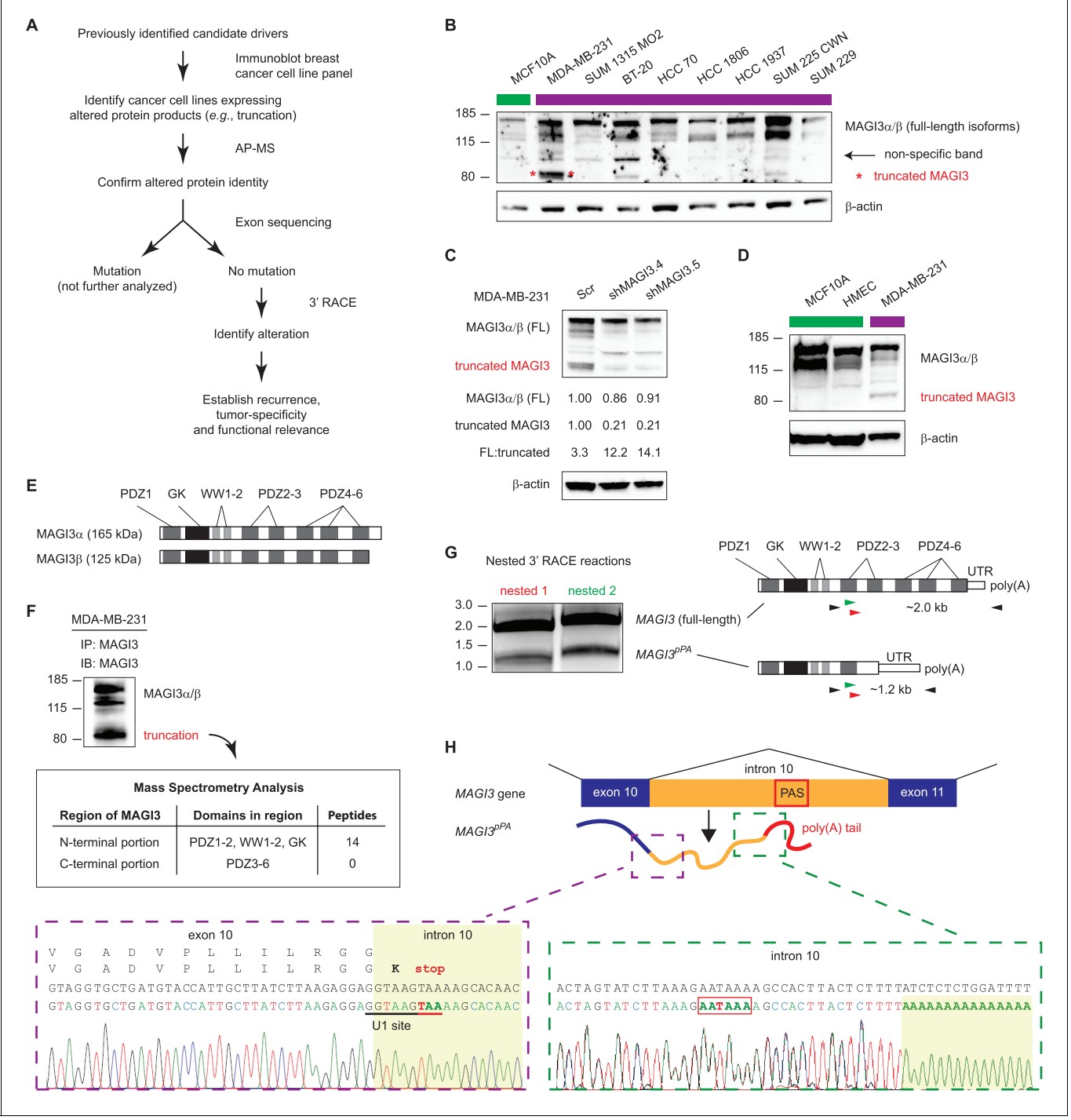

**Figure 1.** Premature polyadenylation of *MAGI3* in the MDA-MB-231 breast cancer cell line causes the expression of a truncated MAGI3 protein. (**A**) Strategy for identifying protein products altered by previously overlooked mechanisms. Protein products of previously identified candidate driver genes (*Ni et al., 2013*) were screened by immunoblotting. Altered products were identified by electrophoretic mobility shift and confirmed by affinity purification-mass spectrometry (AP-MS). Exons of corresponding genes were sequenced. Genes not affected by coding-region DNA mutations were analyzed by 3' RACE and sequencing to identify the nature of the altered product and the associated mechanism of alteration. Alteration events caused by mechanisms not widely appreciated in cancer were subsequently investigated for tumor-specific recurrence and functional relevance. (**B**) Immunoblot of *MAGI3* gene products in the screening panel of human breast cell lines. Non-transformed (green) and transformed (violet) cell lines are indicated. A truncated MAGI3 protein (indicated by red asterisks) is expressed in MDA-MB-231 cells. Immunoblot of β-actin is included to show loading. Approximate molecular mass markers are indicated in kDa. (**C**) Immunoblots of *MAGI3* gene products and β-actin levels in MDA-MB-231 cells transduced with the indicated shRNA constructs. The relative levels of full-length and truncated MAGI3 proteins following RNAi-mediated depletion are *Figure 1 continued on next page*

*Figure 1 continued*

normalized to β-actin levels. The ratios of full-length to truncated MAGI (FL:truncated) are also quantified. (D) Immunoblot of *MAGI3* gene products in the indicated non-transformed cell lines (green) and MDA-MB-231 breast cancer cell line (violet). Immunoblot of β-actin is included to show loading. Approximate molecular mass markers are indicated in kDa. (E) Diagrams of the full-length MAGI3 protein isoforms α and β. Both full-length isoforms possess the indicated arrangement of 6 PDZ domains (dark grey), 2 WW domains (light grey) and a catalytically inactive GK domain with homology to the yeast guanylate kinase (black). In addition, isoform α has an extended C-terminal sequence with no known function. (F) Immunoblot for three MAGI3 protein isoforms immunoprecipitated from MDA-MB-231 cell lysate. The gel slice containing the smallest, truncated MAGI3 protein was analyzed by mass spectrometry, and peptides were mapped exclusively to the N-terminal half of the protein. Approximate molecular mass markers are indicated in kDa. (G) Detection of full-length and truncated *MAGI3* mRNA isoforms by 3′ RACE. Products from two independent nested 3′ RACE reactions performed on MDA-MB-231 RNA were separated by agarose gel electrophoresis. Truncated and full-length *MAGI3* transcripts are indicated. Diagrams show regions targeted for amplification and locations of primary (black) and nested (green/red) primers used for 3′ RACE. (H) Diagram of the truncated and prematurely polyadenylated *MAGI3* mRNA (*MAGI3^pPA*) generated from the *MAGI3* gene locus despite the absence of genetic mutations within the gene. *Green box:* 3′ RACE sequencing results of *MAGI3^pPA* show premature cleavage and polyadenylation (shaded yellow) following a cryptic PAS (outlined in red) in intron 10. The corresponding intron 10 genomic sequence is shown above in black. *Violet box:* Upstream of the pPA event, 3′ RACE sequencing results show a short extension of the open-reading frame into intron 10. The corresponding exon 10 and intron 10 genomic sequence is shown above in black, with the intact U1 snRNA binding site (GGTAAG) underlined in black. Intron 10 is shaded yellow. A stop codon (red underscore) occurs 6 nt following the exon-intron junction. Encoded amino acid sequence is shown in black above the nucleotide sequence (wild-type MAGI3, upper line; MAGI3^pPA, lower line).

The following figure supplements are available for figure 1:

**Figure supplement 1.** Sequencing chromatograms of MDA-MB-231 genomic DNA focusing on *MAGI3* exon 10, intron 10 and exon 11.

**Figure supplement 2.** Sequencing chromatograms of the *MAGI3^pPA* product from exon 10 to the poly(A) tail downstream of the cryptic PAS in intron 10.

Accordingly, we interrogated breast cancer cell lines for evidence of previously unannotated protein products of candidate drivers. Immunoblotting across this cell line panel for one such candidate, MAGI3, revealed a number of extra bands in addition to the two full-length protein isoforms, α and β (*Figure 1B*). We found that one of these bands, faintly appearing in most of the cell lines, was non-specific to MAGI3 since it could not be depleted by multiple shRNA targeting *MAGI3* (*Figure 1C*). However, in MDA-MB-231 breast cancer cells, a strong band of lower molecular weight was observed and could be specifically depleted by RNAi (*Figure 1B and C*). Notably, this truncation was not expressed in the non-transformed MCF10A mammary cell line or *TERT*-immortalized human mammary epithelial cells (HMEC) (*Figure 1B and D*).

MAGI3 is a scaffolding protein with 2 WW and 6 PDZ domains, along with a catalytically inactive region of homology to the yeast guanylate kinase (*Figure 1E*) (*Funke et al., 2005*). Mass spectrometry (MS) analysis of the immunoprecipitated truncation mapped many peptides corresponding to the N-terminal half of MAGI3, but detected no peptides from the PDZ domains 3–6 that consist of the C-terminal half of MAGI3 (*Figure 1F*). To identify the genetic alteration responsible for this truncation, we sequenced all 22 exons and flanking intron sequences of *MAGI3* but found no mutations. This indicated that the truncation is not generated by DNA mutation of the *MAGI3* coding sequence or splice sites. We subsequently used 3′ rapid amplification of cDNA ends (RACE) to isolate the transcript responsible for the truncated MAGI3 protein. This yielded a truncated *MAGI3* mRNA isoform corresponding to the size and mapped regions of the truncated protein (*Figure 1G*). Sequencing the 3′ end of the mRNA revealed a cleavage and polyadenylation event involving a cryptic poly(A) signal (PAS) located in *MAGI3* intron 10 (*Figure 1H* and *Figure 1—figure supplement 1*). Consequently, the normal splicing event from exon 10 to exon 11 of *MAGI3* does not occur in the truncated mRNA (*MAGI3^pPA*), resulting in a stop codon several nucleotides into intron 10 and premature transcription termination following the cryptic intronic PAS (*Figure 1H*).

Local cis-acting DNA mutations in upstream U1 snRNA binding sites or downstream U-rich elements (URE) can affect splicing as well as cleavage and polyadenylation (*Kaida et al., 2010*; *Berg et al., 2012*; *Almada et al., 2013*; *Proudfoot, 2011*). Therefore, we sequenced the entire *MAGI3* intron 10 from MDA-MB-231 cells but were unable to find any mutations (*Figure 1—figure supplement 1*). We also sequenced the entire region of the *MAGI3^pPA* 3′ RACE product corresponding to intron 10 and the flanking exon 10 sequence, but again uncovered no mutations (*Figure 1—*

*figure supplement 2*). Notably, the known cis-acting sequences involved in proper splicing and repression of pPA, including the upstream/downstream splice donors, upstream U1 snRNA binding site and downstream URE, were intact in the sequencing analyses (*Figure 1H* and *Figure 1—figure supplements 1* and *2*). We thus conclude that this pPA event in intron 10 of *MAGI3* arises independently of local genetic mutations. However, we cannot exclude the possibility that a long-range, non-coding mutation in *MAGI3* might play a role in this phenomenon.

## Evidence for recurrent generation of *MAGI3^pPA* in human breast cancer

To determine whether the pPA product of *MAGI3* is physiologically relevant in the context of primary human breast tumors, we analyzed whole transcriptome shotgun sequencing (RNA-Seq) data made available by The Cancer Genome Atlas (TCGA) and a previously published RNA-Seq study (*Varley et al., 2014*). We focused on data collected from primary breast tumors and matched normal tissues in order to establish whether pPA of *MAGI3* is tumor-specific and recurrent. In total, these data included 160 primary breast tumors along with 160 matched, tumor-adjacent normal samples.

Since large differences in *MAGI3* sequencing depth between the two sets of samples might bias detection of pPA events, we first compared sequencing depth of *MAGI3* between tumor and normal RNA-Seq data and found it to be comparable, with only a 1.1-fold median increase in tumor samples. We subsequently used a three-tiered strategy to identify occurrences of *MAGI3^pPA* in the data (*Figure 2A*). To start, we calculated the genome coverage of total mappable reads for each sample and then asked if intronic reads are present in *MAGI3* intron 10. The vast majority of tumor-adjacent normal samples (157 out of 160) had either no intronic reads or a few, isolated reads; only three samples exhibited long stretches of overlapping intronic reads (*Figure 2A*). In comparison, a larger fraction of primary tumors (26 out of 160) were observed with extended stretches of overlapping *MAGI3* intron 10 reads (*Figure 2A*).

Premature polyadenylation and intron retention events can both account for intronic reads in RNA-Seq data. Therefore, our next goal was to rule out any samples with intronic read patterns due to intron retention. While reads unique to pPA events are expected to fall within the pPA-specific intronic interval, reads arising from intron retention should exhibit a more uniform distribution throughout the entire intron. Thus, for intron retention, the expected distribution of reads upstream and downstream of the cryptic PAS in *MAGI3* intron 10 can be compared to the empirical read distributions in the 29 samples with tracts of overlapping intronic reads (26 tumor, 3 normal) (*Figure 2A*). Interestingly, seventeen of these samples (16 tumor, 1 normal) showed read distributions that were inconsistent with intron retention ($p < 0.05$, $\chi^2$ test) (*Figure 2A*).

Next, we evaluated the evidence for *MAGI3^pPA* expression in these seventeen samples by examining the continuity of read coverage across the pPA-specific region (*Figure 2A*). In this analysis, we defined continuity as having overlapping read coverages across the pPA-specific interval with four or fewer small (<75 nt) gaps. This eliminated five samples (4 tumor, 1 normal) that had either larger gaps or more than four small gaps in coverage (*Figure 2A*). We considered the remaining twelve tumor samples, displaying continuous read signals across the pPA region (*Figure 2B*), to express the *MAGI3^pPA* isoform. The relative expression levels of the truncated isoform was between 2% to 6% of *MAGI3α* expression levels, but was generally higher than or comparable to *MAGI3β* expression levels (*Figure 2D*). The occurrence of the *MAGI3* pPA event in 7.5% (12 out of 160) of primary breast tumors and the absence of this event in matched tumor-adjacent normal tissues (0 out of 160) suggest that it is a tumor-specific and recurrent event that may play an important role in breast cancer pathogenesis (*Figure 2C*).

## MAGI3^pPA and other breast cancer-associated *MAGI3* truncations contribute to malignant transformation

To determine whether pPA at the *MAGI3* locus in human breast cancers might affect features of malignancy, we evaluated the role of *MAGI3^pPA* on anchorage-independent growth (*Figure 3A and B*). We took advantage different RNAi sequences targeting *MAGI3* that varied in their efficacy in knocking down levels of the full-length protein. Using two independent *MAGI3*-targeting RNAi sequences (*Figure 3C*), we were able to largely deplete MAGI3^pPA in MDA-MB-231 cells while not significantly altering levels of full-length MAGI3 (*Figure 1C*). This caused a greater than four-fold

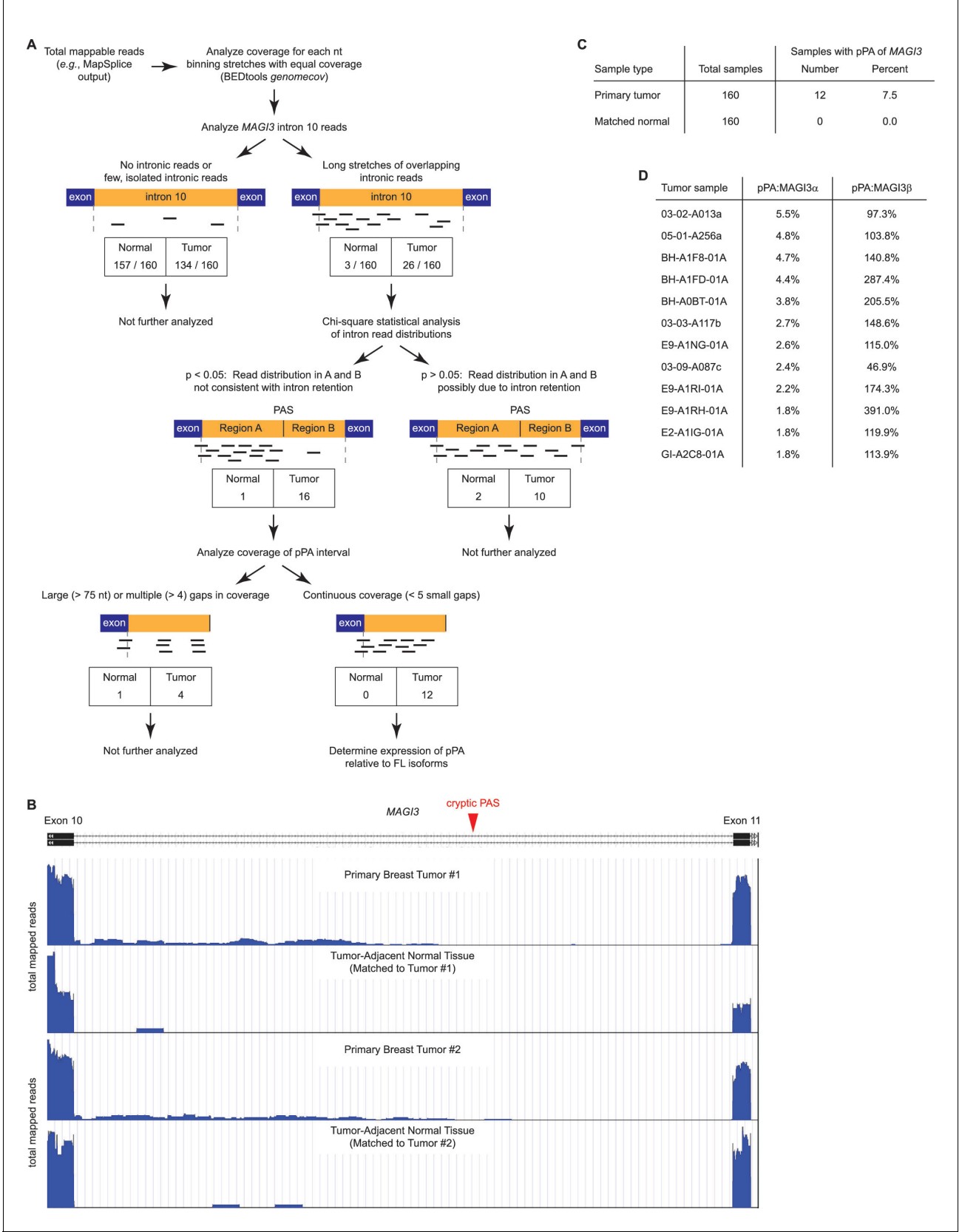

**Figure 2.** Premature polyadenylation of *MAGI3* is a recurrent, tumor-specific event in breast cancer. (**A**) Summary of RNA-Seq analysis, statistical tests, and thresholds used to identify primary tumor and matched, tumor-adjacent normal samples expressing *MAGI3^{pPA}*. Genome coverage of total

*Figure 2 continued on next page*

*Figure 2 continued*

mappable reads were quantified for each sample, and read coverage across the tenth intron of *MAGI3* was analyzed. Samples with sustained stretches of overlapping reads were identified and statistically analyzed for read patterns consistent with intron retention using a $\chi^2$ test. Samples with read patterns in which the intron retention null hypothesis was rejected (p<0.05) were further analyzed for continuous read coverage across the pPA-specific region. Samples with large gaps (>75 nt) or multiple gaps (>4 gaps) failed to pass the threshold set for *MAGI3^pPA*. (B) Total read coverage (blue) of *MAGI3* intron 10 in two primary breast tumors and matched, tumor-adjacent normal samples. The location of the cryptic PAS is indicated by the red arrowhead. (C) Table summary of the percentage of primary breast cancers or matched normal tissues in which premature polyadenylation of *MAGI3* occurs. (D) Relative expression levels of *MAGI3^pPA* versus full-length *MAGI3α* and *MAGI3β* isoforms in primary breast tumor samples.

increase in the relative expression levels of full-length MAGI3 versus MAGI3^pPA (FL:truncated) (*Figure 1C*). Intriguingly, the loss of MAGI3^pPA impaired the growth of MDA-MB-231 cells on soft agar compared to non-silencing Scr control, as shown by the reduced size and number of colonies (p=1.1 x 10^{-4} and 3.4 x 10^{-5} for shMAGI3.4 and shMAGI3.5, respectively) (*Figure 3A and B*). These data suggest that the *MAGI3* intron 10 pPA event may have been positively selected by MDA-MB-231 cells during the process of malignant transformation, and thus implicate pPA as a mechanism of alteration capable of contributing to breast cancer.

To corroborate these findings, we further asked whether other *MAGI3* truncations found in breast cancer – generated by known alteration mechanisms – also exhibit oncogenic properties. A *MAGI3-AKT3* fusion gene, caused by a chromosomal rearrangement involving breakpoints at *MAGI3* intron 9 and *AKT3* intron 1, has been previously identified in breast cancer (*Banerji et al., 2012*). To examine the consequences of this *MAGI3* truncation on its own, we created a corresponding *MAGI3* construct truncated at exon 9 and fused it to GFP (MAGI3ΔC-GFP) (*Figure 3D*). Compared to MAGI3^pPA, which partially retains the PDZ3 domain, MAGI3ΔC-GFP lacks all residues corresponding to PDZ2 and PDZ3 (*Figure 3E*). When ectopically expressed along with the SV40 early region in MCF10A cells (MCF10A-SV40), MAGI3ΔC-GFP significantly promoted anchorage-independent growth on soft agar compared to the poor growth of control MCF10A-SV40 cells expressing GFP (p=4.8 x 10^{-5}) (*Figure 3F and G*). Together, these results suggest that truncated *MAGI3* products, whether generated by pPA or other alteration mechanisms, promote mammary cell transformation.

## Full-length MAGI3 interacts with the Hippo effector YAP and inhibits YAP-dependent transformation

To understand how truncated *MAGI3* causes malignant transformation, we first investigated the function of wild-type *MAGI3*. Because the kinase domain of MAGI3 lacks catalytic activity (*Funke et al., 2005*), we focused on identifying MAGI3 protein interactors by affinity purification of V5-tagged, full-length MAGI3 in MCF10A cells followed by mass spectrometry (AP-MS). The YAP oncoprotein – a transcription co-activator that functions as a major effector of the Hippo tumor suppressor pathway – was identified to be top candidate interactor with MAGI3 (*Figure 4A*). SERPINB5, also known as the mammary tumor suppressor Maspin, was also identified as a likely MAGI3 interactor (*Figure 4A*). In addition, several tight junction and cellular adhesion molecules were identified by this analysis (*Figure 4A* and *Supplementary file 1*), validating the AP-MS approach since MAGI3 has been observed to localize to cell junctions (*Adamsky et al., 2003*; *Funke et al., 2005*). Indeed, the subcellular localization of MAGI3 in human breast epithelium is consistent with these previous reports, with tight junction localization in luminal cells and cytoplasmic localization in basal/myoepithelial cells (*Figure 4—figure supplement 1*).

Since the YAP oncoprotein was by far the top interacting protein with MAGI3 (*Supplementary file 1*), we performed reciprocal co-immunoprecipitation (co-IP) experiments for endogenous MAGI3 and endogenous YAP to validate their interaction. Indeed, we found that YAP co-immunoprecipitated with MAGI3 and likewise, MAGI3 co-immunoprecipitated with YAP from MCF10A cell lysates (*Figure 4B*). We next determined the critical domains necessary for interaction between MAGI3 and YAP by ectopically expressing various epitope-tagged full-length and deletion mutants of MAGI3 and YAP in HEK293T cells. To determine whether this interaction required PDZ domain-binding, we performed co-IP experiments with V5-tagged MAGI3 and either Flag-tagged full-length YAP or a YAP mutant lacking the PDZ-binding motif (YAPΔC) (*Oka et al., 2010*). Consistent with the endogenous co-IP results, Flag-tagged full-length YAP co-immunoprecipitated with V5-

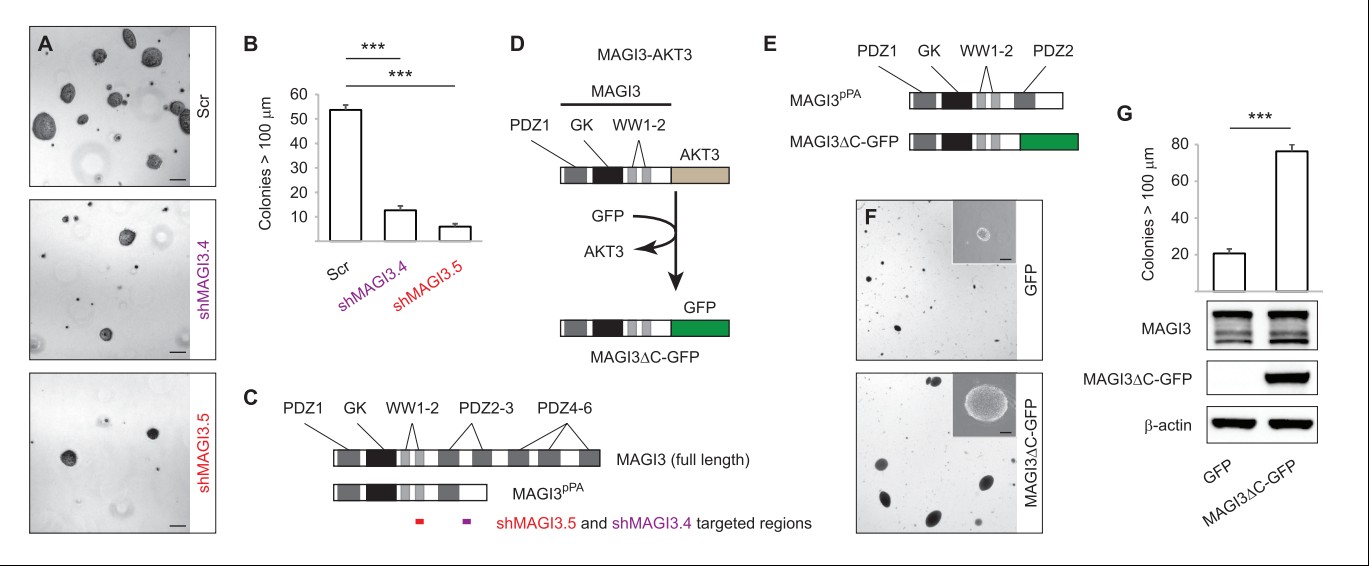

**Figure 3.** *MAGI3^pPA* and a pPA-independent, breast cancer-associated *MAGI3* truncation promote anchorage-independent growth. (**A**) Anchorage-independent growth assays for MDA-MB-231 cells expressing Scr control or two independent shMAGI3 (shMAGI3.4 and shMAGI3.5) sequences that preferentially deplete *MAGI3^pPA*. Images show representative soft agar fields for the indicated cell lines after two weeks. Scale bar represents 100 µm. (**B**) Quantification of soft agar colonies in the indicated cell lines (n = 3 biological replicates per cell line) after two weeks. Data are presented as mean ± SEM. ***p<0.001 (two-tailed Student's t-tests). (**C**) Diagrams of full-length MAGI3 and MAGI3^pPA showing the regions targeted by shMAGI3.4 and shMAGI3.5. (**D**) Diagrams of the protein domains of a breast cancer-associated MAGI3 truncation, fused to GFP (MAGI3ΔC-GFP), corresponding to the *MAGI3-AKT3* product caused by a chromosome 1 inversion event identified by whole-genome sequencing (*Banerji et al., 2012*). (**E**) Diagrams of the protein domains of the MAGI3^pPA and MAGI3ΔC-GFP truncations. (**F**) Anchorage-independent growth assays for MCF10A-SV40 cells expressing MAGI3ΔC-GFP or GFP control. Images show representative soft agar fields and insets show commonly observed colony sizes for the indicated cell lines after three weeks. Scale bar represents 100 µm. (**G**) Quantification of transformed colonies in the indicated cell lines (n = 3 biological replicates per cell line) after three weeks (upper panel). Immunoblots of MAGI3ΔC-GFP, full-length MAGI3 isoforms, and β-actin levels in the cell lines (lower panels). Data are presented as mean ± SEM. ***p<0.001 (two-tailed Student's t-test).

tagged MAGI3 in HEK293T cells (*Figure 4C*). However, Flag-YAPΔC and MAGI3-V5 could not be co-immunoprecipitated by either anti-Flag or anti-V5 IP (*Figure 4C*), indicating that the PDZ-binding motif of YAP is necessary for the physical interaction between the two proteins. To further confirm that the interaction between YAP and MAGI3 requires PDZ domain-binding, we first utilized a computational predictor to map the PDZ-binding peptide of YAP against all human PDZ domains (*Hui and Bader, 2010*) and found that the sixth PDZ domain (PDZ6) of MAGI3 was the highest-scoring interactor on the list (*Supplementary file 2*). We thus expressed a Flag-tagged *MAGI3* construct lacking PDZ6 (MAGI3ΔPDZ6) in HEK293T cells and performed co-IP experiments for YAP to determine the necessity of PDZ6 for MAGI3 and YAP interaction. Deletion of PDZ6 curtailed MAGI3 interaction with YAP (*Figure 4D*). Taken together, these data demonstrate that the PDZ-binding motif of YAP and the PDZ6 domain of MAGI3 are necessary for protein-protein interaction.

Having shown that full-length MAGI3 and YAP interact endogenously, we next investigated whether manipulating MAGI3 levels affects the ability of YAP to act as a transcription co-activator. When MCF10A cells were depleted of *MAGI3* by RNAi, the expression of YAP target genes *ANKRD1* and *CYR61* were induced (p=9.9 x 10$^{-3}$ and 2.5 x 10$^{-4}$, respectively), as determined by quantitative real-time PCR (qPCR) (*Figure 4E*). The rescue of *MAGI3* levels restored the expression of *ANKRD1* and *CYR61* to baseline levels (p=0.012 and 3.5 x 10$^{-4}$, respectively) (*Figure 4E*), thus demonstrating the specificity of YAP regulation by *MAGI3*. We next developed a phenotypic assay to determine whether the molecular interactions between MAGI3 and YAP are physiologically relevant. Because RNAi-mediated knockdown of *MAGI3* in MCF10A-SV40 cells causes anchorage-independent growth (p=1.4 x 10$^{-3}$) (*Figure 4F and G*), and overexpression of YAP in MCF10A cells also causes transformation (*Overholtzer et al., 2006*), we used the transformed phenotype to conduct epistasis analysis between *MAGI3* and *YAP1*. Knockdown of *YAP1* in *MAGI3*-depleted cells restored

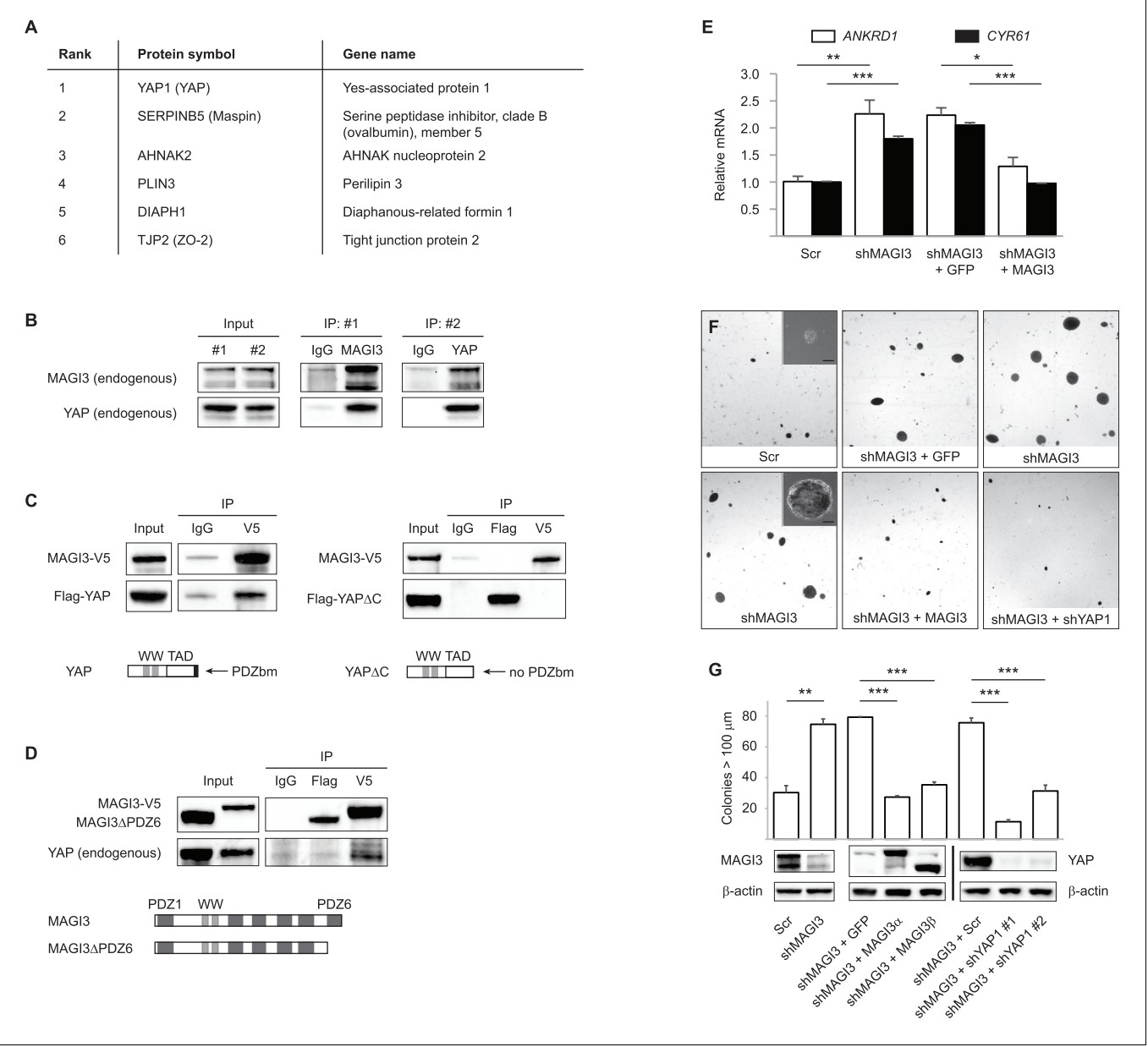

**Figure 4.** MAGI3 interacts with the YAP oncoprotein and suppresses YAP-dependent transformation. (**A**) List of the top candidate MAGI3-interacting proteins in MCF10A cells as identified by AP-MS. (**B**) Endogenous MAGI3/YAP interaction in MCF10A cells by co-IP. Immunoblots of MAGI3 and YAP from input and indicated IP lysates are shown. (**C**) Immunoblots of full-length MAGI3-V5 and either full-length Flag-YAP (left) or mutant Flag-YAPΔC (right), in which the C-terminal YAP PDZ-binding motif (FLTWL) is deleted, from the indicated input and IP lysates (HEK293T cells). (**D**) Immunoblots of full-length YAP and either full-length MAGI3-V5 or PDZ6-deleted MAGI3-V5 from the indicated input and IP lysates (HEK293T cells). (**E**) The relative expression of YAP target genes *ANKRD1* and *CYR61* in MCF10A cells, as determined by qPCR, in cells expressing shMAGI3 or non-silencing Scr control, with or without *MAGI3* cDNA or non-rescuing GFP control (n = 3 technical replicates per cell line). Data are presented as mean ± SEM. *p<0.05, **p<0.01, ***p<0.001 (two-tailed Student's t-tests). (**F**) Anchorage-independent growth assays for MCF10A-SV40 cells expressing the indicated shRNA and/or cDNA constructs. Images show representative soft agar fields and insets show commonly observed colony sizes for the indicated cell lines after three weeks. Scale bar represents 100 μm. (**G**) Quantification of transformed colonies in the indicated cell lines (n = 3 biological replicates per cell line) after three weeks (upper panel). Immunoblots of full-length MAGI3 isoforms, YAP and β-actin levels in the cell lines (lower panels). Data are presented as mean ± SEM. **p<0.01, ***p<0.001 (two-tailed Student's t-tests).

The following figure supplement is available for figure 4:

**Figure supplement 1.** MAGI3 expression and localization in normal human mammary epithelial cells.

the normal phenotype (p=4.9 x 10$^{-5}$ and 8.9 x 10$^{-4}$ for shYAP1 #1 and #2, respectively), as did restoring *MAGI3* expression in *MAGI3*-depleted cells (p=6.5 x 10$^{-7}$ and 2.0 x 10$^{-5}$ for MAGI3α and MAGI3β, respectively) (*Figure 4F and G*). Collectively, these data suggest that the protein-protein interaction between MAGI3 and YAP is physiologically relevant for the negative regulation of YAP by full-length MAGI3.

## Oncogenic MAGI3 truncations activate YAP by dominantly suppressing full-length MAGI3/YAP interactions

Having obtained an understanding of the relationships between full-length MAGI3, YAP and malignant transformation, we proceeded to investigate the functional differences between truncated and full-length MAGI3. YAP functions as a transcriptional coactivator, and the regulation of its subcellular localization is a major determinant of protein activity (*Zhao et al., 2007*). In HEK293T cells, YAP activity exhibits highly robust responses to serum deprivation, which promotes cytoplasmic accumulation of YAP, and subsequent serum stimulation, which rapidly induces YAP nuclear translocation (*Yu et al., 2012*). Thus, we examined the subcellular localization of YAP in cells transfected with either truncated or full-length MAGI3. Consistent with the *MAGI3* loss-of-function data, overexpression of full-length MAGI3 led to YAP cytoplasmic accumulation despite serum stimulated cell culture, whereas nuclear translocation of YAP was induced in untransfected control cells (*Figure 5A and B*). By contrast, the expression of truncated MAGI3 caused the opposite effect by increasing nuclear YAP localization in cells even under the serum deprived conditions that normally preclude YAP from the nucleus (*Figure 5C and D*). The increased nuclear YAP localization was indicative of heightened YAP activation, since the amount of phosphorylated/inactive YAP was reduced in lysates from cells expressing MAGI3$^{pPA}$ and MAGI3ΔC-GFP compared to GFP control (*Figure 5E*). Thus, in contrast to the inhibitory effect of full-length MAGI3 on YAP, these data demonstrate that truncated MAGI3 is positively associated with YAP activation. We further asked whether the transformed phenotype caused by expression of truncated MAGI3 is YAP-dependent. Indeed, knockdown of YAP in MCF10A-SV40 cells expressing MAGI3ΔC-GFP resulted in a significant reduction of colony formation on soft agar (p=7.4 x 10$^{-5}$) (*Figure 5F and G*), suggesting that the increase in YAP activity caused by truncated MAGI3 is functionally important for transformation.

Since truncated MAGI3 induces YAP activation and anchorage-independent growth, whereas full-length MAGI3 suppresses YAP activity and the transformed phenotype, we hypothesized that the MAGI3 truncation products found in breast cancer possess dominant-negative activity. Indeed, the presence of truncated MAGI3 was sufficient to prevent endogenous, full-length MAGI3 from interacting with YAP (*Figure 6A*). However, because both MAGI3$^{pPA}$ and MAGI3ΔC-GFP lack the PDZ6 domain required for interaction with YAP (*Figure 4D*), neither truncation was able to physically associate with YAP (*Figure 6B and C*). The absence of physical interaction between truncated MAGI3 and YAP demonstrates that truncated MAGI3 proteins do not directly compete with full-length MAGI3 for YAP binding.

We next asked whether truncated MAGI3 proteins might physically associate with full-length MAGI3. By co-immunoprecipitation, we found that truncated and full-length MAGI3 proteins do indeed physically interact (*Figure 6D and E*). These results suggest that the ability of truncated MAGI3 proteins to form complexes with full-length MAGI3 interferes with physiological MAGI3/YAP interactions. To further confirm that truncated MAGI3 behaves in this dominant-negative manner, we expressed full-length *MAGI3* in MDA-MB-231 cells to induce stoichiometric gains of full-length MAGI3 versus MAGI3$^{pPA}$ (FL:pPA). Alongside these cells, we generated FL:pPA ratio-unmodified controls by stably transducing MDA-MB-231 cells with empty vector GFP. Both pools of cells were orthotopically implanted into female NOD/SCID mice, and tumor growth was monitored over a seven-week timespan. Consistent with the mechanism, the growth of tumors expressing high ratios of FL:pPA was significantly impaired compared to FL:pPA ratio-unmodified control tumors (p=5.8 x 10$^{-5}$) (*Figure 6F and G*). Collectively, these data demonstrate that the recurrent *MAGI3*$^{pPA}$ product and other breast cancer-associated MAGI3 truncations suppress the inhibitory interaction between full-length MAGI3 and YAP, thus providing a molecular link between their dominant-negative activity and their oncogenic nature.

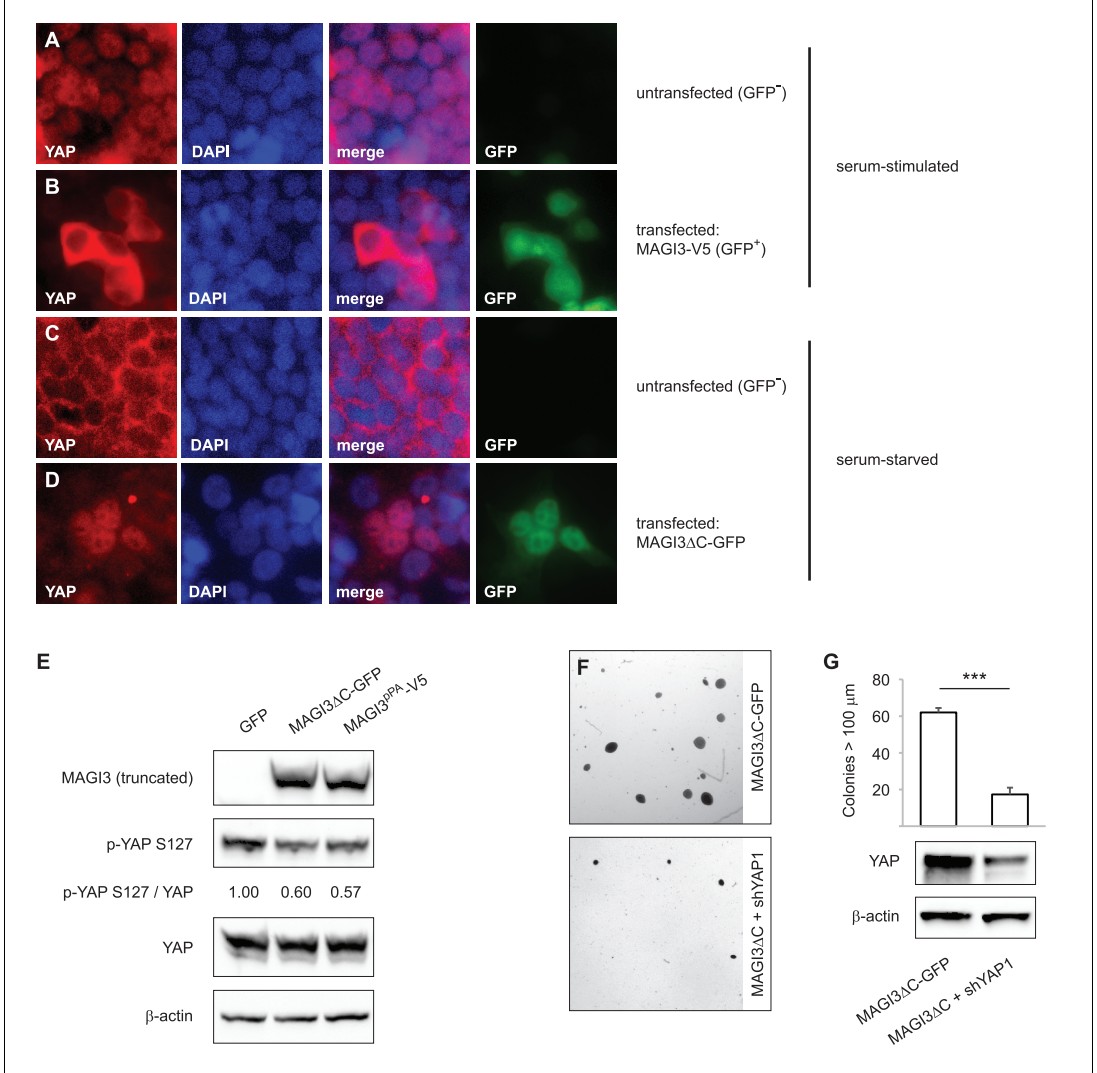

**Figure 5.** Expression of truncated MAGI3 leads to increased YAP activation and YAP-dependent transformation. (**A-B**) Immunofluorescence of serum-stimulated HEK293T cells showing YAP (red) localization. GFP positivity indicates full-length MAGI3-transfected cells. DAPI (blue) was used to visualize nuclei. (**A**) Field containing predominantly untransfected cells is shown. (**B**) Field containing MAGI3-transfected cells is shown. (**C-D**) Immunofluorescence of serum-deprived HEK293T cells showing YAP (red) localization. GFP indicates expression and localization of MAGI3ΔC-GFP. DAPI (blue) was used to visualize nuclei. (**C**) Field containing predominantly untransfected cells is shown. (**D**) Field containing MAGI3ΔC-GFP-transfected cells is shown. (**E**) Immunoblots of V5-tagged MAGI3$^{pPA}$, MAGI3ΔC-GFP, Ser127-phosphorylated YAP, total YAP and β-actin levels in HEK293T cells transfected as indicated. The relative levels of Ser127-phosphorylated YAP in the indicated lysates are normalized to total YAP levels. (**F**) Anchorage-independent growth assays for MCF10A-SV40 cells expressing MAGI3ΔC-GFP with or without shYAP1. Images show representative soft agar fields for the indicated cell lines after three weeks. Scale bar represents 100 μm. (**G**) Quantification of transformed colonies in the indicated cell lines (n = 3 biological replicates per cell line) after three weeks (upper panel). Immunoblots of YAP and β-actin levels in the cell lines (lower panels). Data are presented as mean ± SEM. ***p<0.001 (two-tailed Student's t-tests).

## Discussion

In this study, we have identified a breast cancer-associated pPA event at a cryptic intronic PAS in *MAGI3*, resulting in the production of a truncated, oncogenic gene product. We have shown that truncations of MAGI3 generated by pPA as well as other mechanisms of alteration promote increased activation of the Hippo effector YAP and induce YAP-dependent malignant transformation. These effects are mediated by the dominant-negative activity of MAGI3 truncations, which physically associate with full-length MAGI3 and antagonize the inhibitory protein-protein interaction between full-length MAGI3 and YAP. Thus, the interaction between full-length MAGI3 and YAP plays

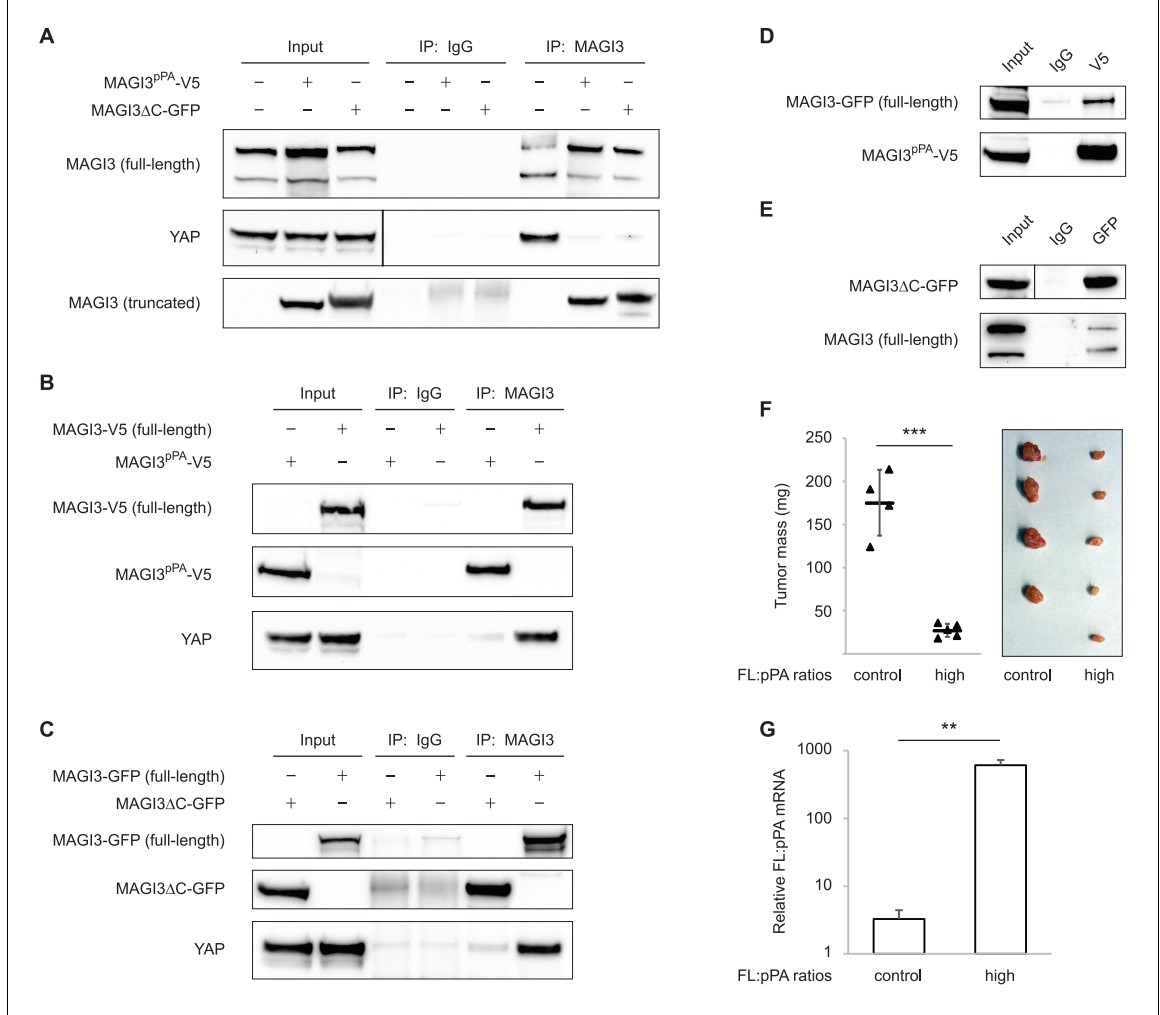

**Figure 6.** Dominant-negative MAGI3 truncations interact with full-length MAGI3 proteins to interfere with MAGI3/YAP interaction. (**A**) Immunoblots of full-length endogenous MAGI3, endogenous YAP, V5-tagged MAGI3$^{pPA}$ and MAGI3ΔC-GFP from the indicated input and IP lysates (HEK293T cells). (**B**) Immunoblots of V5-tagged MAGI3$^{pPA}$, full-length MAGI3-V5 and endogenous YAP from the indicated input and IP lysates (HEK293T cells). (**C**) Immunoblots of MAGI3ΔC-GFP, full-length MAGI3-GFP and endogenous YAP from the indicated input and IP lysates (HEK293T cells). (**D**) Immunoblots of full-length MAGI3-GFP and V5-tagged MAGI3$^{pPA}$ from the indicated input and IP lysates (HEK293T cells). (**E**) Immunoblots of MAGI3ΔC-GFP and endogenous full-length MAGI3 from the indicated input and IP lysates (HEK293T cells). (**F**) Quantification of tumor mass (left panel) and photograph (right panel) of MDA-MB-231 cells grown as orthotopic xenografts in NOD/SCID mice. Tumors with high ratios of FL:pPA MAGI3 (n = 5 mice) or unmodified control ratios (n = 4 mice) are indicated. Data are plotted as individual data points along with mean ± SD. *** p<0.001 (two-tailed Student's t-tests). (**G**) The relative expression ratios of FL:pPA mRNA in control- (n = 4) or high-ratio (n = 5) MDA-MB-231 tumor xenografts, as determined by qPCR. Data are presented as mean ± SEM. ** p<0.01 (two-tailed Student's t-test).

a central role in suppressing malignant transformation (*Figure 7*). Furthermore, our identification and functional characterization of *MAGI3$^{pPA}$* establishes that pPA can contribute to cancer by utilizing cryptic intronic PAS to generate truncated mRNA isoforms encoding oncoproteins.

## The tumor suppressive role of MAGI3/YAP interactions

In the past, a number of observations have supported the possibility that *MAGI3* may be involved in cancer. For example, oncogenic human papillomaviruses, which have evolved to target critical tumor suppressors necessary for transformation such as p53, also target MAGI3 for degradation (*Thomas et al., 2002*). In addition, *MAGI3* was found to be part of a downregulated gene signature associated with breast tumor-initiating cells (*Creighton et al., 2009*). However, a specific role for

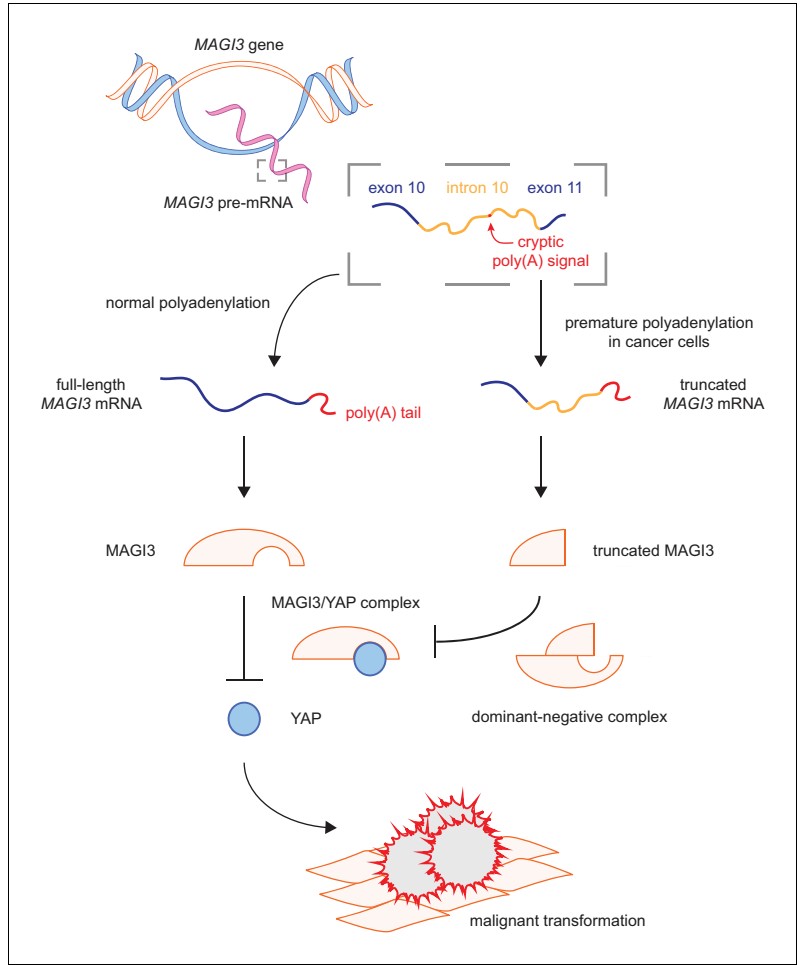

**Figure 7.** Model for the mechanism by which premature polyadenylation of *MAGI3* contributes to malignant transformation. In normal cells, the proper post-transcriptional processing of transcripts produced by the *MAGI3* gene locus gives rise to full-length MAGI3 proteins that negatively regulate the YAP oncoprotein by forming an inhibitory complex. In cancer cells, premature polyadenylation of *MAGI3* results in the production of truncated MAGI3 proteins in addition to full-length MAGI3. Truncated MAGI3 is unable to bind YAP but can interact with full-length MAGI3. This dominant-negative interaction may prevent full-length MAGI3 from forming an inhibitory complex with YAP, thus resulting in YAP-mediated malignant transformation.

*MAGI3* as a tumor suppressor in human cancer has remained elusive, due to the apparent infrequent alteration of the gene in cancer and the lack of cancer-relevant mechanistic evidence.

In human cancer, *MAGI3* is not frequently altered by nonsynonymous mutations or copy number losses (*Cerami et al., 2012*; *Gao et al., 2013*), and has not been previously nominated as a driver. Here, we have found evidence that a single alteration event in *MAGI3*, caused by the utilization of a cryptic intron 10 PAS by premature cleavage and polyadenylation, occurs in approximately 7.5% of 160 breast cancers analyzed, and the resulting truncated *MAGI3* product contributes to the malignant transformation of human mammary epithelial cells. Transformation caused by an independent MAGI3 truncation, also found in breast cancer, provides additional functional evidence for the cancer relevance of MAGI3 truncations. Moreover, the ability of these truncations to function as dominant-negative proteins and prevent the inhibition of YAP by full-length MAGI3 suggests that the generation of truncated *MAGI3* products by pPA or chromosomal rearrangement may more potently deregulate YAP activity and drive transformation compared to the mutation or loss of a single *MAGI3* allele. Collectively, these data indicate that *MAGI3* alterations in cancer may occur much more frequently than previously appreciated and support the clinical relevance of MAGI3 truncations as driver alterations.

By defining the protein-protein interaction network of MAGI3, we have uncovered the previously unknown interaction between MAGI3 and YAP, an oncoprotein that represents the major downstream effector of the Hippo tumor suppressor pathway (*Pan, 2010*; *Johnson and Halder, 2014*). We have demonstrated that these YAP-inhibitory interactions play an important role in suppressing malignant transformation, since reduction of these functional complexes by depleting full-length MAGI3 leads to increased YAP-dependent transcriptional activation and transformation. Importantly, we found that the C-terminal PDZ6 domain of MAGI3 is essential for MAGI3/YAP interaction, whereas the two independent breast cancer-associated MAGI3 truncations studied here – MAGI3$^{pPA}$ and MAGI3ΔC – lack this domain and fail to interact with YAP. Nevertheless, the truncations dominantly suppressed interactions between full-length MAGI3 and YAP, resulting in YAP activation and malignant transformation. We further determined that MAGI3 truncations can physically associate with full-length MAGI3 proteins, and thus posit that this aberrant complex interferes with the interaction between full-length MAGI3 and YAP. Our results underscore the importance of the MAGI3/YAP interaction in suppressing malignant transformation and implicate MAGI3 as a new potential component involved in the regulation of the Hippo tumor suppressor pathway. In addition, our identification of other candidate protein-protein interactors of MAGI3 establishes a resource for future investigation into other biological processes regulated by full-length MAGI3 protein.

## Premature cleavage and polyadenylation as a cancer-relevant alteration mechanism

An emerging phenomenon in cancer is the generation of gene expression alterations by localized mutational or transcriptome-wide events affecting cleavage and polyadenylation. In mantle cell lymphoma, for example, mutations that generate novel, gene-proximal PAS in the *CCND1* 3′ UTR have been discovered. In these tumors, the production of shorter, prematurely polyadenylated cyclin D1 mRNA isoforms, which exhibit dramatically extended half-lives, drive the overexpression of cyclin D1 oncoproteins (*Wiestner et al., 2007*). In addition, a number of studies on genes with tandem PAS arrangements in the 3′ UTR have shown that shorter 3′ UTRs utilizing the gene-proximal PAS are favored by cancer cells (*Sandberg et al., 2008*; *Singh et al., 2009*; *Mayr and Bartel, 2009*; *Xia et al., 2014*). This increased tendency in cancer cells to utilize gene-proximal PAS for cleavage and polyadenylation corresponds to increased mRNA half-life as well as protein expression (*Singh et al., 2009*; *Mayr and Bartel, 2009*), suggesting that 3′ UTR shortening may be a general mechanism of evading regulation by tumor suppressive microRNAs.

Our study extends these previous findings by showing that, in cancer, premature cleavage and polyadenylation can occur in the intron of a gene despite the absence of mutations in the immediate exon/intron sequences flanking the cryptic PAS. While alternative 3′ UTR PAS selection does not alter the mRNA coding potential, pPA can cause gene-truncating alterations that produce pre-mRNAs with potential to encode constitutively active or dominant-negative proteins. Such pre-mRNAs are naturally able to evade post-transcriptional quality-control surveillance mechanisms such as nonsense-mediated mRNA decay. By producing constitutively active or dominant-negative proteins, pPA transcripts may cause significant functional and phenotypic consequences even if expressed at lower levels than wild-type mRNA isoforms. Determining whether pPA events in cancer occur in other genes beside *MAGI3*, and whether pPA is triggered widely throughout the transcriptome or focally restricted to susceptible intronic PAS, are important future directions but beyond the scope of the current study. However, pPA truncation of other cancer-relevant genes is a strong possibility considering the previous studies that have identified dominant-negative proteins arising from pPA events. Prominent examples of these pPA-generated dominant-negative proteins include an EPRS isoform in peripheral blood monocytes that blocks translational repression by full-length EPRS following γ-interferon activation (*Yao et al., 2012*), and a VEGFR2 truncation that inhibits HUVEC tube formation in vitro (*Vorlová et al., 2011*). Given these precedents, it will be important to conduct transcriptome-scale studies to identify other genes affected by premature polyadenylation in cancer. Such studies, however, must await the development of custom pPA-finding computational methods.

The results of our study also pose several intriguing mechanistic questions. For example, how do pPA events arise independently of local, cis-acting genetic mutations? Previous work has shown that, under certain conditions, pPA events can be induced in cells. For example, U1 snRNP function is essential for protecting pre-mRNAs from pPA at cryptic intronic PAS (*Kaida et al., 2010*;

*Almada et al., 2013*), and experimental depletion of U1 snRNP levels can lead to the production of novel, pPA-truncated isoforms that encode dominant-negative proteins (*Vorlová et al., 2011*). Over-expression of polyadenylation factors has also been shown to mediate alternative intronic PAS usage (*Takagaki et al., 1996*). These findings raise the possibility that alteration of a *trans*-acting factor in cancer cells could result in either focal or transcriptome-wide pPA. Moreover, because the relative abundance of U1 snRNP to nuclear pre-mRNAs is a critical factor for determining whether global pPA events occur (*Berg et al., 2012*), alterations that drive global transcriptional amplification, a phenomenon frequently observed in cancer cells (*Nie et al., 2012*; *Lin et al., 2012*; *Sabò et al., 2014*; *Cunningham et al., 2014*), might also induce pPA. In addition, whether broad pPA induction enables cancer cells to sample and select specific pPA events that confer functional growth advantages – such as *MAGI3^{pPA}* – and how pPA events at the functionally relevant loci are passed from mother cell to daughter cells, are outstanding questions that will require further investigation into the phenomenon of pPA in cancer.

## Materials and methods

### Cloning and plasmid construction

Full-length *MAGI3* isoforms concordant with NCBI RefSeq transcripts were amplified from MCF10A cDNA and subcloned into pLenti7.3-V5-DEST (Thermo Fisher Scientific, Waltham, MA) or CS-CG (*Miyoshi et al., 1998*) for MAGI3-V5 and MAGI3-GFP, respectively. Truncated *MAGI3* isoforms were amplified from using internal, sequence-specific primers. MAGI3ΔPDZ6 was subcloned into pcDNA3.0-Flag-HA (Addgene, Cambridge, MA). MAGI3ΔC was subcloned into CS-CG to generate MAGI3ΔC-GFP. MAGI3^{pPA} was subcloned into pLenti7.3-V5-DEST. shMAGI3 (TRCN0000037865), shMAGI3.4 (TRCN0000037867) and shMAGI3.5 (TRCN0000037868) were subcloned into pLKO.1-puro. *YAP1* shRNAs (TRCN0000107265 and TRCN0000107267) were subcloned into pLKO.1-hygro. Expression vectors for SV40 early region, Scr shRNA, Flag-YAP, and Flag-YAPΔC have been described previously (*Oka et al., 2008*; *Oka et al., 2010*; *Wu et al., 2009*; *Ni et al., 2013*).

### Cell lines and tissue culture

Cell lines used in this study (MDA-MB-231, MCF10A, BT-20, HCC 70, HCC 1806, HCC 1937 and HEK293T) were purchased from ATCC (Manassas, VA), which have been authenticated by short tandem repeat DNA profiling and are free of mycoplasma contamination. Other cell lines were obtained from Dr. Stephen Ethier (SUM lines) or established in our laboratory (HMEC). Cells were grown at 37°C with 5% $CO_2$ and cultured in DMEM supplemented with 10% fetal bovine serum unless otherwise noted. MCF10A cells were cultured in complete MEGM (Lonza, Switzerland) plus 100 ng/ml cholera toxin. HMECs were cultured in complete MEGM (Lonza). SUM lines were cultured in Ham's F12 supplemented with 5% fetal bovine serum, 5 µg/ml insulin and 0.5 µg/ml hydrocortisone. For serum starvation, cells were incubated for 24 hr in media without supplements. Subsequent serum stimulation was performed by adding FBS to a final concentration of 10% and incubation for 30 m. Transfections were performed using FuGENE HD (Promega, Madison, WI). For lentivirus production, HEK293T cells were co-transfected with pCMV-VSV-G, pCMV-ΔR8.2-Δvpr, and lentiviral expression vectors. For the generation of cell lines, MCF10A and MDA-MB-231 cells were incubated overnight in viral supernatants supplemented with 5 µg/ml protamine sulfate and subsequently selected by antibiotics or FACS. Cell lines harboring multiple genetic manipulations were created by serial transductions.

### RNA isolation and downstream applications

Total RNA was isolated using the RNeasy Mini Kit (Qiagen, Germany) and treated with DNase I (Roche, Switzerland) to remove any contaminating DNA. For 3′ RACE, cDNA was synthesized by reverse transcription using Superscript II RTase and an oligo(dT)-containing adapter primer to poly (A) mRNA (Thermo Fisher Scientific). Subsequently, a nested PCR method was used to amplify *MAGI3* transcripts. The primary reaction utilized a *MAGI3*-specific forward primer (M3Q2-F) and a unique amplification primer (UAP-R) to the oligo(dT)-containing adapter primer, and the secondary reaction utilized *MAGI3*-specific forward primers downstream of M3Q2-F (M3-F12 and M3-F13). The

**Table 1.** Primer sequences used for 3′ RACE.

| Name | Primer Sequence (5′ to 3′) |
|---|---|
| M3Q2-F | CTGTGTCCTCGGTCACACTC |
| M3-F12 | TATCCATGGCATCGTCAGGC |
| M3-F13 | GTTGCTGCTACCCCTGTCAT |
| UAP-R | GGCCACGCGTCGACTAGTAC |

3′ ends of resulting products were sequenced by Sanger sequencing. Primer sequences used for 3′ RACE are listed in *Table 1*.

For real-time PCR (qPCR), cDNA was first synthesized from total RNA by reverse transcription using the iScript cDNA synthesis kit (Biorad, Hercules, CA). Samples were diluted for reactions with gene-specific primers and SYBR green (Bioline, Taunton, MA) using the CFX96 Real-Time System (Biorad). qPCR was performed in triplicate for each sample-target combination, and mRNA abundance was normalized to *GAPDH*. Primer sequences are listed in *Table 2*.

## Protein isolation and downstream applications

For immunoprecipitation experiments, cells were suspended in Tris-HCl buffer (20 mM Tris-HCl, pH 7.5) supplemented with protease/phosphatase inhibitors, lysed by dounce homogenizer, pre-cleared by incubating with Protein A Dynabeads (Thermo Fisher Scientific) for 1 hr, then incubated with antibodies overnight. Complexes were then bound to Protein A Dynabeads for 1 hr, washed three times with Tris-HCl buffer and eluted by boiling in SDS loading buffer. To isolate protein from whole-cell lysates for immunoblotting, cells were lysed in RIPA buffer (50 mM Tris-HCl, pH 8.0, 150 mM NaCl, 0.5% sodium deoxycholate, 0.1% SDS, 1% NP-40) supplemented with protease inhibitors (Roche) phosphatase inhibitors (Sigma-Aldrich, St. Louis, MO). Protein samples were separated by SDS-PAGE according to standard procedures, transferred onto nitrocellulose membranes and blocked with 5% milk. Immunoblotting was performed according to standard procedures and protein detection was visualized using enhanced chemiluminescence (Thermo Fisher Scientific). For protein mass spectrometry, MAGI3 IP lysates were separated by SDS-PAGE, fixed in the gel, stained with a 0.3% Coomassie Blue R250 solution, then destained overnight. The gel slice containing the truncated MAGI3 protein from MDA-MB-231 cells was excised, digested and analyzed by liquid chromatography-tandem mass spectrometry (Taplin Mass Spectrometry Facility, Harvard Medical School). For identification of MAGI3-interacting proteins from MCF10A cells, the gel was partitioned into six sections for digestion and downstream analysis. The accepted list of interacting proteins was obtained by subtracting common contaminants (CRAPome). Antibodies used in the above applications are listed in *Table 3*.

## Immunofluorescence

Disease-free human breast tissue was obtained in compliance with the laws and institutional guidelines as approved by the Tufts Medical Center Institutional Review Board Committee. The approval number for human subject research is 00004517. The tissue was obtained from patients undergoing elective reduction mammoplasty. De-identified breast tissue was utilized for this study, and for this

**Table 2.** Primer sequences used for qPCR.

| Gene | Forward Primer (5′ to 3′) | Reverse Primer (5′ to 3′) |
|---|---|---|
| ANKRD1 | AGTAGAGGAACTGGTCACTGG | TGGGCTAGAAGTGTCTTCAGAT |
| CYR61 | GGTCAAAGTTACCGGGCAGT | GGAGGCATCGAATCCCAGC |
| GAPDH | CCATGGGGAAGGTGAAGGTC | TAAAAGCAGCCCTGGTGACC |
| MAGI3 (FL) | AGCAGTTTCCAGTAGGTGCT | TGTCGTCCTCGGGTTGTTTT |
| MAGI3 (pPA) | AGCAGTTTCCAGTAGGTGCT | GAGGTCCAGATGACACACCA |

**Table 3.** Antibodies used in this study.

| Antibody | Host | Vendor/catalog number | IB | IP | IF |
|---|---|---|---|---|---|
| β-actin | Mouse | Abcam (Cambridge, MA) ab6276 | | | |
| Flag (HRP-) | Rabbit | Cell Signaling Technology (Danvers, MA) 2044 | X | | |
| Flag | Mouse | Sigma-Aldrich F3165 | | X | |
| GFP | Rabbit | Abcam ab290 | X | X | |
| KRT14 | Mouse | Vector Laboratories (Burlingame, CA) VP-C410 | | | X |
| MAGI3 | Mouse | Abcam ab11509 | X | | |
| MAGI3 | Rabbit | Novus Biologicals (Littleton, CO) NBP2-17210 | X | X | X |
| mouse (Alexa488-) | Goat | Thermo Fisher Scientific A11001 | | | X |
| mouse (HRP-) | Goat | Cell Signaling Technology 7076 | X | | |
| normal IgG | Mouse | Santa Cruz Biotechnology (Dallas, TX) sc-2025 | | X | |
| normal IgG | Rabbit | Santa Cruz Biotechnology sc-2027 | | X | |
| rabbit (Alexa546-) | Goat | Thermo Fisher Scientific A11010 | | | X |
| rabbit (HRP-) | Goat | Cell Signaling Technology 7074 | X | | |
| V5 (HRP-) | Rabbit | Abcam ab1325 | X | | |
| V5 | Mouse | Thermo Fisher Scientific R960 | | X | X |
| YAP | Mouse | Santa Cruz Biotechnology sc-101199 | X | X | X |
| YAP | Rabbit | Santa Cruz Biotechnology sc-15047 | X | | X |
| p-YAP Ser 127 | Rabbit | Cell Signaling Technology 4911 | X | | |
| ZO-1 | Mouse | T 339100 | | | X |

reason informed consent was not required. For paraffin-embedded human mammary tissues, sections were deparaffinized and rehydrated prior to heat-induced antigen retrieval in Tris-EDTA (10 mM Tris, pH 9.0, 1 mM EDTA, 0.05% Tween 20). Cultured cells seeded on coverslips were fixed in 4% paraformaldehyde and permeabilized in 0.1% Triton-X 100 in PBS. Samples were blocked in 3% normal goat serum for 1 hr at room temperature prior to incubation with primary antibodies overnight. Alexa488- and Alexa546-conjugated secondary antibodies were used to detect proteins and 1 μg/ml DAPI was used to visualize nuclei. Samples were mounted on glass slides with the Slow-Fade Antifade Kit (Thermo Fisher Scientific). Antibodies used for immunofluorescence are listed in *Table 3*.

## Transformation and orthotopic tumor xenograft assays

For soft agar transformation assays, $10^4$ cells were suspended in a 0.4% Seaplaque agarose:culture media mixture and layered onto 0.8% agarose in DMEM. Fresh growth medium was provided weekly. Colonies were imaged, stained with 0.005% crystal violet and quantified (colonies $\geq$ 100 μm diameter) after 14 days (MDA-MB-231) or 21 days (MCF10A). For orthotopic tumor xenograft assays, all procedures were performed in accordance with the animal protocol approved by the Tufts University Institutional Animal Care and Use Committee. The approval number for animal research is A-3775-01. Prior to surgery, 10-week old, female NOD/SCID mice (The Jackson Laboratory, Bar Harbor, ME) were anesthetized by isoflurane vapor. An incision was made along the right flank to expose the inguinal mammary gland, and 1 x $10^6$ cells in a total volume of 25 μl 1:1 Matrigel:phosphate-buffered saline were injected into the gland. Post-operative analgesic and monitoring were provided. Animals were sacrificed after 7 weeks, and tumors were dissected and weighed.

## Bioinformatic analysis of RNA-seq data

Sequence alignment data and normalized gene expression data for primary breast tumors and matched, tumor-adjacent normal tissues were downloaded from the Cancer Genomics Hub, TCGA data portal, the NCBI Sequence Read Archive (SRA) and the Gene Expression Omnibus (GEO). The identifiers for the SRA and GEO datasets are SRP042620 and GSE58135 (*Varley et al., 2014*). To

evaluate sequencing depth for *MAGI3* in primary tumor and tumor-adjacent normal tissues of the two datasets, we compared upper quartile-normalized TPM (transcripts per million) or FPKM (fragments per kilobase of transcript per million mapped reads) values for *MAGI3.* To identify putative pPA events, we calculated the genome coverage of total mappable reads for each sample by using the BEDtools *genomecov* script and converted the output to BEDGRAPH files for subsequent analyses using the UCSC genome browser. Next, we inspected each sample for evidence of read coverage across *MAGI3* intron 10, allowing us to group samples into two categories: (1) samples with no intronic reads or only a few, isolated intronic reads; (2) samples with sustained stretches of overlapping intronic reads. The intronic read distributions of samples in Group 2 were further analyzed by a $\chi^2$ goodness-of-fit test to evaluate the null hypothesis—namely, that the observed distribution of sequencing reads occurs due to intron retention. This was performed by splitting *MAGI3* intron 10 into two regions: (A) the pPA-specific region from the start of the intron to the cryptic PAS; (B) the intronic region downstream of the cryptic PAS. A $p<0.05$ was considered a significant deviation from the expected read distribution and therefore a rejection of the null hypothesis. Samples with $p<0.05$ were further analyzed for continuity of read coverage throughout Region A. To account for non-uniformity biases inherent in RNA-Seq read distributions, we set a threshold for continuity defined by overlapping read coverage with fewer than 5 small (<75 nt) gaps. Samples passing all the above criteria were considered to express the pPA-truncated *MAGI3* isoform, and relative expression levels of $MAGI3^{pPA}$ versus either of the full-length isoforms ($MAGI3\alpha$ and $MAGI3\beta$) were measured by counting the number of reads mapping uniquely to each isoform and dividing the length-corrected reads mapping to the pPA isoform by those mapping to each full-length isoform.

## Statistical analysis

A power analysis program for comparing two means (two-sample, two-sided equality) was used to determine the sample size needed for experiments, with significance criteria being $\alpha = 0.05$ and power = 0.8. For these calculations, we assumed a difference between means of 30 and 50 along with standard deviations of 10 and 20 for transformation assays and tumor xenograft experiments, respectively. Data were analyzed and compared between groups using two-tailed Student's t-tests. A $p<0.05$ was considered statistically significant.

## Acknowledgements

We thank S Kwok and A Parmelee for performing FACS; R Tomaino and the Taplin Mass Spectrometry Facility at Harvard Medical School for mass spectrometry analysis; J Elman for technical assistance; A Vilborg for insightful discussions regarding premature transcription termination; and W Zhou and J Li for critical reading of the manuscript. This work was supported by a grant from the American Cancer Society #PF-14-046-01-DMC (to TKN), funding from the Raymond & Beverly Sackler Convergence Laboratory (to CK) and grants from ArtBeCAUSE, the Breast Cancer Research Foundation, and the NIH/NCI CA170851 and NIH/NICHD HD073035 (to CK).

## Additional information

### Funding

| Funder | Grant reference number | Author |
| --- | --- | --- |
| American Cancer Society | Postdoctoral Fellowship, PF-14-046-01-DMC | Thomas K Ni |
| Breast Cancer Research Foundation | | Charlotte Kuperwasser |
| National Institutes of Health | R01 CA170851 | Charlotte Kuperwasser |
| ArtBeCAUSE | | Charlotte Kuperwasser |
| Raymond & Beverly Sackler Foundation | | Charlotte Kuperwasser |
| National Institutes of Health | R01 HD073035 | Charlotte Kuperwasser |

The funders had no role in study design, data collection and interpretation, or the decision to submit the work for publication.

## Author contributions
TKN, Conceptualisation, Methodology, Investigation, Writing, Funding Acquisition, Acquisition of data, Analysis and interpretation of data; CK, Conceptualisation, Writing, Funding Acquisition, Supervision

## Author ORCIDs
Thomas K Ni, http://orcid.org/0000-0003-2961-0593
Charlotte Kuperwasser, http://orcid.org/0000-0001-7913-9619

## Ethics
Human subjects: Disease-free human breast tissue was obtained in compliance with the laws and institutional guidelines as approved by the Tufts Medical Center Institutional Review Board Committee. The approval number for human subject research is 00004517. The tissue was obtained from patients undergoing elective reduction mammoplasty. De-identified breast tissue was utilized for this study, and for this reason informed consent was not required.

Animal experimentation: This study was performed in accordance with the recommendations in the Guide for the Care and Use of Laboratory Animals of the National Institutes of Health. All animals were handled according to the animal protocol approved by the Tufts University Institutional Animal Care and Use Committee. The approval number for animal research is A-3775-01.

## Additional files
### Supplementary files
• Supplementary file 1. Comprehensive list of candidate MAGI3-interacting proteins in MCF10A cells.

• Supplementary file 2. Top human PDZ domains predicted to bind the YAP PDZ-binding motif.

### Major datasets
The following previously published datasets were used:

| Author(s) | Year | Dataset title | Dataset URL | Database, license, and accessibility information |
|---|---|---|---|---|
| Varley KE, Gertz J, Roberts BS, Davis NS, Bowling KM, Kirby MK, Nesmith AS, Oliver PG, Grizzle WE, Forero A, Buchsbaum DJ, LoBuglio AF, Myers RM | 2014 | Breast Cancer RNA-Seq | http://www.ncbi.nlm.nih.gov/geo/query/acc.cgi?acc=GSE58135 | Publicly available at NCBI Gene Expression Omnibus (accession no: GSE58135) |
| The Cancer Genome Atlas (TCGA) | 2016 | Breast Invasive Carcinoma RNA-Seq | http://www.ncbi.nlm.nih.gov/biosample/?term=phs000178 | Publicly available at NCBI BioSample (accession no: phs000178) |
| Varley KE, Gertz J, Roberts BS, Davis NS, Bowling KM, Kirby MK, Nesmith AS, Oliver PG, Grizzle WE, Forero A, Buchsbaum DJ, LoBuglio AF, Myers RM | 2014 | Breast Cancer RNA-Seq | http://www.ncbi.nlm.nih.gov/sra/?term=SRP042620 | Publicly available at NCBI Sequence Read Archive (accession no: SRP042620) |

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
