## [Decision Letter]

Thank you for submitting your article "Premature Polyadenylation and Truncation of *MAGI3* Contribute to Transformation and Breast Cancer" for consideration by *eLife*. Your article has been favorably evaluated by Sean Morrison (Senior Editor) and three reviewers, one of whom is a member of our Board of Reviewing Editors.

The reviewers have discussed the reviews with one another and the Reviewing Editor has drafted this decision. *eLife* is open to reviewing a revised version of the paper, though substantial new data will be required to address the reviewer comments.

Summary:

The authors have identified a C-terminal truncated form of MAGI3 protein as a cancer-promoting variant. Interestingly, the novel form of MAGI3 results from translation of a product of premature cleavage and polyadenylation (pPA) of the transcript at a cryptic intronic poly(A) signal. The truncated form of MAGI3 acts as a dominant-negative to inhibit a newly-discovered function of MAGI3 as an inhibitor of YAP. The paper is interesting both for this example of the role of pPA in cancer and for the new insights into YAP regulation. However, the reviewers have raised several major concerns. Substantial new data will be needed to address the reviewers' comments. Since the new data will mostly come from further in-depth analysis of sequencing data, together with a few new experiments, we are offering you the opportunity to modify the paper and resubmit. However, if you cannot provide the data within the two months we normally allow for return of a revised submission, you should write back to indicate how you plan to proceed and indicate the length of time you expect it would take to comply with the required additional work. Alternatively, we understand that you may feel it is not possible to comply in which case you may choose to withdraw and submit this work elsewhere.

Essential revisions:

1) The cause of the pPA event in MAGI3 is unknown, and may be due to a cis-acting mutation within the *MAGI3* gene or to a broad effect on many genes. The authors emphasize that the pPA is not due to mutation. To address this: (a) the entire *MAGI3* gene, not just the exons, must be sequenced; (b) the data from MDA-MB-231 cells and other cells with pPA of *MAGI3*, as well as some pPA-negative lines (e.g. MCF10A), should be mined for the occurrence of pPA events in other genes; and (c) the RNASeq data needs to be presented more clearly, statistical methods explained, and the relative abundance of RNA and protein isoforms in normal and cancer cells determined.

2) The prevalence of pPA of *MAGI3* in breast cancer is unclear. Analysis of TCGA data or other published data sets is required.

3) The literature on pPA in cancer should be discussed.

The comments of each reviewer are provided below for your information. The major comments have been distilled into the above essential revisions.

*Reviewer #1:*

1) It should be possible to access more RNASeq data to get better statistics on the frequency of tumors in which magi3 RNA is truncated.

2) Better data on YAP subcellular localization should be provided.

3) The authors should discuss previous evidence for premature cleavage/polyadenylation in cancer in more depth. In addition to the Mayr and Bartel 2009 paper, other seminal papers are Sandberg et al. 2008 Science 320:1643 and Singh et al. 2009 Cancer Res 69:9422. Previous examples of dominant negatives arising through transcript truncation include Yao et al. 2012 Cell 149:88 and Vorlova et al. 2011 Mol Cell 43:927.

*Reviewer #2:*

1) The authors strongly emphasize that they searched for abnormal proteins that were /not/ the result of genetic changes. However, it is unclear whether they have, in fact, ruled out the possibility that intronic mutations are responsible for the truncated MAGI3 isoform. They state: "We sequenced MAGI3 but found no mutations in coding regions or exon-flanking intron sequences, indicating that this truncation was not generated by DNA mutation."

However, as far as I can tell, that they didn't sequence the entire intron containing the cryptic poly(A) site. Therefore, they can't conclude that local intronic mutations aren't responsible for the observed truncation. (Conversely, it's formally possible that a trans-acting cleavage or poly(A) factor could also be mutated. This possibility is difficult to rule out.) The bottom line is that the authors should not claim so strongly that they are studying unusual proteins that don't result from mutations.

2) The analyses illustrated in Figure 2 are neither well-described nor statistically robust. It's unclear whether the data shown in Figure 2 is normalized w.r.t. total read coverage, which is an obvious potential confounding factor. Normalized read coverage plots are the relevant visual illustration. More importantly, the statistical thresholds that the authors used are insufficiently described. The description in "Bioinformatic Analysis of RNA-Seq Data" is not detailed enough to assess whether the authors are correctly distinguishing between premature polyadenylation and intron retention, etc. For this kind of analysis – where they're comparing occurrence of an unusual isoform between primary cancers and peritumoral normal – using a robust statistical procedure is essential.

3) The authors need to clearly measure and describe the relative quantitative levels of the truncated vs. full-length isoforms of MAGI3. For example, the authors should have a single Western blot against MAGI3 showing their positive (MDA-MB-231) as well as negative (MCF10A, HMEC) samples (currently in Figure 1). Similarly, the authors should quantitatively measure relative expression levels of the truncated isoform in their analyses of primary samples (see comment #2 above). Quantitative comparisons are important because it's relatively rare for an aberrant isoform to be completely absent from normal cells, but highly expressed in cancer cells. Quantitative differences are much more common.

4) The authors overemphasize the novelty of their results. For example, they state in the Abstract that: "Our results suggest that pPA, by generating truncated and oncogenic mRNA isoforms, contributes to cancer as a previously overlooked mechanism of alteration."

Similarly, they state in the Discussion that: "Premature cleavage and polyadenylation has not previously been known to alter genes in cancer."

However, alternative polyadenylation, including premature polyadenylation, has been extensively studied in the literature. For example, gene-proximal poly(A) sites are preferentially used in cancers (e.g., Sandberg et al., Science, 2008). Other authors have also studied truncated isoforms of specific cancer-relevant genes (e.g., Wiestner et al., Blood, 2007). Unusual isoforms generated by alternative splicing and premature polyadenylation have been particularly well-studied in the context of ligand-independent isoforms of the androgen receptor (e.g., Antonarakis et al., NEJM, 2014).

*Reviewer #3:*

1) The authors used 3' RACE to implicate premature polyadenylation of MAGI3 in just one breast cancer cell line (MDA-MB-231). It would be more convincing if the same analysis can be done in the three primary tumors which the authors believe have similar aberration based on RNA-seq. This will exclude the possibility that the premature polyadenylation is something that happens during in vitro culture of cancer cell lines.

2) The RNA-seq data in Figure 2 is incomplete. It does not show coverage after the cryptic polyadenylation site. It is important to show coverage through the whole *MAGI3* gene, including all exons and introns, especially the ones after the cryptic polyadenylation site.

3) The study left open the question of how such premature polyadenylation happens. Although the underlying molecular mechanisms may be the subject of a future study), it might be informative to analyze the RNA-seq data at whole genome level for the three primary tumors to determine whether premature polyadenylation happens in other genes. At least this will tell us whether this is a local or genome-wide phenomenon.

4) In Figure 6, the authors showed xenografts of MDA-MB-231 cells expressing full-length MAGI3. The expression level of MAGI3 full-length vs. truncated form should be examined to show their stoichiometry.

[Editors' note: further revisions were requested prior to acceptance, as described below.]

Thank you for resubmitting your work entitled "Premature Polyadenylation of MAGI3 Produces a Novel Driver of Human Breast Cancer" for further consideration at *eLife*. Your article has been favorably evaluated by Sean Morrison (Senior Editor) and three reviewers, one of whom is a member of our Board of Reviewing Editors.

The manuscript has been improved but there are some remaining issues that need to be addressed before acceptance, as outlined below:

1) The authors need to clarify whether the RNA-seq data that they analyze in Figure 2 is controlled for sequencing depth and expression of the *MAGI3* gene. If MAGI3 is much more highly expressed in cancer than in peritumoral normal cells, then they will a priori be much more likely to detect rare premature polyadenylation in the tumors than in the normals. This concern was raised in the previous review.

2) The authors still overstate the novelty of their results in the Abstract, where they state: "Here we report a new mechanism also capable of altering gene products to produce cancer drivers." This is incongruous with their Discussion, which places their work in the broader context of the field more appropriately. The discovery of premature polyadenylation in cancer is not new. The novelty is in the specific molecular mechanism of YAP regulation, not in pPA. The title is also too strong. Stating that a gene product is a "driver" of cancer typically means that it is sufficient transform cells in vivo, which the authors have not shown. One solution may be to substitute "dominantly-acting oncogene" for "driver" in the title, and replace the second sentence of the abstract with "Here we report a new mechanism by which premature polyadenylation can produce an oncogenic protein."

---

## [Author Response]

Essential revisions:

1) The cause of the pPA event in MAGI3 is unknown, and may be due to a cis-acting mutation within the MAGI3 gene or to a broad effect on many genes. The authors emphasize that the pPA is not due to mutation. To address this: (a) the entire MAGI3 gene, not just the exons, must be sequenced; (b) the data from MDA-MB-231 cells and other cells with pPA of MAGI3, as well as some pPA-negative lines (e.g. MCF10A), should be mined for the occurrence of pPA events in other genes; and (c) the RNASeq data needs to be presented more clearly, statistical methods explained, and the relative abundance of RNA and protein isoforms in normal and cancer cells determined.

2) The prevalence of pPA of MAGI3 in breast cancer is unclear. Analysis of TCGA data or other published data sets is required.

3) The literature on pPA in cancer should be discussed.

The comments of each reviewer are provided below for your information. The major comments have been distilled into the above essential revisions.

Reviewer #1:

1) It should be possible to access more RNASeq data to get better statistics on the frequency of tumors in which magi3 RNA is truncated.

We thank the reviewers for these suggestions and have performed analysis on the full set of paired breast tumor-normal RNA-Seq data from TCGA. This analysis shows a clear picture of *MAGI3^pPA^* in breast cancer (Figure 2). Overall, 12 out of 160 (7.5%) tumors show evidence of pPA truncation of MAGI3 at intron 10 (Figure 2). By contrast, intronic reads in MAGI3 intron 10 from peritumoral normal samples were limited to mostly rare events (Figure 2), and none of these samples showed any evidence of *MAGI3^pPA^* (Figure 2). These findings in the larger TCGA dataset are consistent with the frequency reported in our original manuscript (6%) that was based on a smaller non-TCGA dataset.

In light of these significant findings for tumor-recurring premature polyadenylation of MAGI3 in TCGA data, we felt that the importance of the pPA isoform – as a novel driver isoform of breast cancer – should be conveyed by the title of the paper. As such, we have revised the title of the manuscript to reflect this (now entitled “Premature Polyadenylation of MAGI3 Produces a Novel Driver of Human Breast Cancer”).

2) Better data on YAP subcellular localization should be provided.

We have provided better YAP subcellular localization images in Figure 5.

3) The authors should discuss previous evidence for premature cleavage/polyadenylation in cancer in more depth. In addition to the Mayr and Bartel 2009 paper, other seminal papers are Sandberg et al. 2008 Science 320:1643 and Singh et al. 2009 Cancer Res 69:9422. Previous examples of dominant negatives arising through transcript truncation include Yao et al. 2012 Cell 149:88 and Vorlova et al. 2011 Mol Cell 43:927.

We thank the reviewers for pointing out these omissions. In the revised Discussion section of the manuscript we discuss in greater depth the previous evidence for alternative 3’ UTR polyadenylation in cancer and previous examples for dominant-negative proteins arising from pPA truncation (entitled “Premature Cleavage and Polyadenylation as a Cancer-Relevant Alteration Mechanism”).

Reviewer #2:

1) The authors strongly emphasize that they searched for abnormal proteins that were /not/ the result of genetic changes. However, it is unclear whether they have, in fact, ruled out the possibility that intronic mutations are responsible for the truncated MAGI3 isoform. They state: "We sequenced MAGI3 but found no mutations in coding regions or exon-flanking intron sequences, indicating that this truncation was not generated by DNA mutation."

However, as far as I can tell, that they didn't sequence the entire intron containing the cryptic poly(A) site. Therefore, they can't conclude that local intronic mutations aren't responsible for the observed truncation. (Conversely, it's formally possible that a trans-acting cleavage or poly(A) factor could also be mutated. This possibility is difficult to rule out.) The bottom line is that the authors should not claim so strongly that they are studying unusual proteins that don't result from mutations.

We appreciate the reviewers’ comments and agree that it is important to rule out the presence of local intron 10 mutations in MAGI3. Because the *MAGI3* gene is extremely large (~300 kb), we hope that the reviewers will agree with us that sequencing its entirety would be quite costly and provide little further value in interpreting the results. However, to address this important issue more directly, we have now sequenced the entire intron 10 of MAGI3 and found no mutations. In addition, we have sequenced the entire intron 10 region, along with the preceding exon 10, present in the MAGI3 pPA product isolated by 3’ RACE. This also uncovered no mutations. Both analyses confirm that the major cis-acting sequence features known to dictate proper splicing and repression of cryptic intronic polyadenylation (5’/3’ splice sites, upstream U1 snRNA binding sites, downstream U-rich elements) are not mutated. These data have been added to the Results section and Figure 1—figure supplement 1 and Figure 1—figure supplement 2.

We also agree that is difficult to rule out the presence of mutations in trans-acting cleavage and polyadenylation factors. Likewise, while we have sequenced all 22 exons of MAGI3 and found no mutations, it would be difficult to rule out (or interpret the significance of) long-range, non-coding mutations elsewhere in the very large 300-kb *MAGI3* gene locus. Thus, we have revised our statements regarding the absence of mutations to indicate the absence of mutations in the local exonic and intronic regions surrounding the cryptic PAS in intron 10, since these are supported by our findings.

As the reviewer suggested, we also discuss the possibility of mutation/alteration to a trans-acting cleavage and polyadenylation factor in the revised Discussion section.

2) The analyses illustrated in Figure 2 are neither well-described nor statistically robust. It's unclear whether the data shown in Figure 2 is normalized w.r.t. total read coverage, which is an obvious potential confounding factor. Normalized read coverage plots are the relevant visual illustration. More importantly, the statistical thresholds that the authors used are insufficiently described. The description in "Bioinformatic Analysis of RNA-Seq Data" is not detailed enough to assess whether the authors are correctly distinguishing between premature polyadenylation and intron retention, etc. For this kind of analysis – where they're comparing occurrence of an unusual isoform between primary cancers and peritumoral normal – using a robust statistical procedure is essential.

3) The authors need to clearly measure and describe the relative quantitative levels of the truncated vs. full-length isoforms of MAGI3. For example, the authors should have a single Western blot against MAGI3 showing their positive (MDA-MB-231) as well as negative (MCF10A, HMEC) samples (currently in Figure 1). Similarly, the authors should quantitatively measure relative expression levels of the truncated isoform in their analyses of primary samples (see comment #2 above). Quantitative comparisons are important because it's relatively rare for an aberrant isoform to be completely absent from normal cells, but highly expressed in cancer cells. Quantitative differences are much more common.

We appreciate the comments and suggestions for the RNA-Seq analysis. We have revised the text to provide detailed descriptions of the RNA-Seq analysis scripts/tools, statistical methods, and thresholds used (Results section 2 and Materials and methods). To further improve overall clarity, we have also added visual depictions of our analyses (Figure 2) and included a running tally of tumor and matched normal samples at each level of analysis (Figure 2). As suggested, we also provide read coverage plots for the entire intron 10 of MAGI3, including the region after the cryptic PAS, for tumor-normal pairs (Figure 2).

We thank the reviewers for their suggestions to quantify the stoichiometry/relative expression levels of FL isoforms versus the pPA isoform. Throughout the revision, we provide measurements for the relative expression of the pPA-truncated MAGI3 isoform versus the FL isoforms.

Figure 2: For all 12 primary tumors identified by RNA-Seq analysis to express the pPA isoform of MAGI3, we have quantified the expression levels relative to both full-length MAGI3 isoforms.

Figure 3: We quantify the levels of full-length versus pPA-truncated protein isoforms of MAGI3 in our control and knockdown experiments.

Figure 6: For the tumor xenografts, we use qPCR to quantitatively determine the stoichiometry of FL:pPA MAGI3.

Figure 1: As suggested, we performed a single Western blot against MAGI3 for MDA-MB-231 as well as normal (MCF10A and HMEC) samples. We were unable to detect the pPA-truncated MAGI3 in whole cell lysates obtained from normal cell lines (MCF10A, HMEC), even with longer exposures and higher total protein loaded for these samples compared to MDA-MB-231 cells. We have thus reproduced our original results, reaffirming our conclusion that the pPA truncation is not expressed in the normal mammary cell lines.

4) The authors overemphasize the novelty of their results. For example, they state in the Abstract that: "Our results suggest that pPA, by generating truncated and oncogenic mRNA isoforms, contributes to cancer as a previously overlooked mechanism of alteration."

Similarly, they state in the Discussion that: "Premature cleavage and polyadenylation has not previously been known to alter genes in cancer."

*However, alternative polyadenylation, including premature polyadenylation, has been extensively studied in the literature. For example, gene-proximal poly(A) sites are preferentially used in cancers (e.g., Sandberg et al., Science, 2008). Other authors have also studied truncated isoforms of specific cancer-relevant genes (e.g., Wiestner et al., Blood, 2007). Unusual isoforms generated by alternative splicing and premature polyadenylation have been particularly well-studied in the context of ligand-independent isoforms of the androgen receptor (e.g., Antonarakis et al., NEJM, 2014).*

Please see response to Reviewer 1, point 3.

Reviewer #3:

1) The authors used 3' RACE to implicate premature polyadenylation of MAGI3 in just one breast cancer cell line (MDA-MB-231). It would be more convincing if the same analysis can be done in the three primary tumors which the authors believe have similar aberration based on RNA-seq. This will exclude the possibility that the premature polyadenylation is something that happens during in vitro culture of cancer cell lines.

We agree that this would be a useful analysis, but it requires unavailable raw material from primary tumors. In addition, given the genetic heterogeneity of tumors and the complexity of tumor subtype in breast cancer, analysis of any random 3 primary tumors may not actually be that informative. Thus, to address this issue, we analyzed a larger, paired tumor-normal dataset from TCGA (that consists of >100 primary tumors) and identified additional patient breast cancers that express *MAGI3^pPA^*. Also, we probed multiple breast cancer cell lines for MAGI3 truncation (Figure 1), and the pPA truncation of MAGI3 does not appear to be a frequent event in cultured cancer cell lines suggesting this is not an artifact of ex vivo adaptation.

2) The RNA-seq data in Figure 2 is incomplete. It does not show coverage after the cryptic polyadenylation site. It is important to show coverage through the whole MAGI3 gene, including all exons and introns, especially the ones after the cryptic polyadenylation site.

Please see response to Reviewer 2, points 2 and 3.

3) The study left open the question of how such premature polyadenylation happens. Although the underlying molecular mechanisms may be the subject of a future study), it might be informative to analyze the RNA-seq data at whole genome level for the three primary tumors to determine whether premature polyadenylation happens in other genes. At least this will tell us whether this is a local or genome-wide phenomenon.

We agree with the reviewers that determining the nature of premature polyadenylation in cancer – as a focal or transcriptome-wide phenomenon – is important. However, bioinformatics methods for identifying novel pPA events on a transcriptome-wide scale are currently lacking. In addition, methodology-building studies similar in scale to the proposed study represent major works in their own right (such as Vaquero-Garcia et al., e*Life*, 2016, and Xia et al., Nat. Commun., 2014). Thus, examining other genes for premature polyadenylation will have to await the development of pPA-finding computational methods, which we are currently working towards and is beyond the scope of this paper.

However, in this revised manuscript we focused our efforts on (1) improving our RNA-Seq analyses for identifying the *MAGI3^pPA^* event, and (2) ascertaining the frequency of pPA truncation of MAGI3 in a larger dataset of tumor-normal paired breast cancers from TCGA. In addition, given the importance of these findings, in the revised manuscript we devote a lengthy discussion of the potential impact of studying pPA events in cancer.

References:

Vaquero-Garcia et al. (2016). A new view of transcriptome complexity and regulation through the lens of local splicing variations. *eLife* 5, e11752.

Xia et al. (2014). Dynamic analyses of alternative polyadenylation from RNA-Seq reveal a 3’-UTR landscape across seven tumour types. Nat. Commun. 5, 5274.

4) In Figure 6, the authors showed xenografts of MDA-MB-231 cells expressing full-length MAGI3. The expression level of MAGI3 full-length vs. truncated form should be examined to show their stoichiometry.

Please see response to Reviewer 2, points 2 and 3.

[Editors' note: further revisions were requested prior to acceptance, as described below.]

The manuscript has been improved but there are some remaining issues that need to be addressed before acceptance, as outlined below:

1) The authors need to clarify whether the RNA-seq data that they analyze in Figure 2 is controlled for sequencing depth and expression of the MAGI3 gene. If MAGI3 is much more highly expressed in cancer than in peritumoral normal cells, then they will a priori be much more likely to detect rare premature polyadenylation in the tumors than in the normals. This concern was raised in the previous review.

We apologize for this omission. We compared upper quartile-normalized TPM or FPKM values for MAGI3 between tumor and tumor-adjacent normal tissues. The relative *MAGI3* expression is only slightly higher in tumors versus adjacent normal tissues (median ~ 1.1-fold increase), and this modest increase in sequencing depth of MAGI3 in tumor samples is unlikely to bias detection of premature polyadenylation of MAGI3. The methods and results are now reported in the manuscript.

*2) The authors still overstate the novelty of their results in the Abstract, where they state: "Here we report a new mechanism also capable of altering gene products to produce cancer drivers." This is incongruous with their Discussion, which places their work in the broader context of the field more appropriately. The discovery of premature polyadenylation in cancer is not new. The novelty is in the specific molecular mechanism of YAP regulation, not in pPA. The title is also too strong. Stating that a gene product is a "driver" of cancer typically means that it is sufficient transform cells in vivo, which the authors have not shown. One solution may be to substitute "dominantly-acting oncogene" for "driver" in the title, and replace the second sentence of the abstract with "Here we report a new mechanism by which premature polyadenylation can produce an oncogenic protein."*

Thank you for the helpful suggestions. We have made these changes to the title and Abstract of the manuscript.